# TARGETED MILP INSTANCE GENERATION VIA FORMULATION CODE RETRIEVAL

## ABSTRACT

Efficient and controllable data generation is critical for improving the performance of data-driven Mixed-Integer Linear Programming (MILP) solvers, especially in applications facing data scarcity. However, existing MILP instance generation methods typically require training a separate model for each problem class, which can be computationally intensive and does not allow for the generation of instances with varying sizes and solution difficulties. To address these challenges, we introduce MILP-Retrieval, a framework for targeted MILP instance generation via formulation code retrieval. We first build a diverse MILP library that includes multiple modalities and use it to pretrain an MILP embedding model. Based on the output of this embedding model, we propose a novel similarity metric that accurately measures the similarity between instances of different sizes within the same problem class. MILP-Retrieval leverages this new metric to retrieve the formulation code of a target instance and further tune it. Experimental results demonstrate the effectiveness of generating MILP instances through formulation code retrieval, with the ability to control both the scale and difficulty of the generated instances. This approach provides a novel perspective on MILP instance generation and opens up new possibilities for learning-based solvers.

## 1    INTRODUCTION

Mixed-Integer Linear Programming (MILP) is widely used in various domains, such as scheduling (Caumond et al., 2009; Floudas & Lin, 2005), logistics (Song et al., 2018; Galvez et al., 2015), and planning (Ren & Gao, 2010). Recently, learning-based solvers (Li et al., 2024; Wang et al., 2023; Ye et al., 2023) have shown promising performance, surpassing traditional solvers (Gurobi Optimization, LLC, 2024; Bolusani et al., 2024; Holmström et al., 2009), offering new opportunities to efficiently tackle complex MILP problems. However, a key challenge in developing learning-based MILP solvers is the scarcity of high-quality data (Gleixner et al., 2021; Bengio et al., 2021). Unlike fields such as natural language processing or computer vision, where large-scale datasets are readily available (Dubey et al., 2024), MILP lacks publicly available, diverse instance datasets. This shortage has led to growing interest in MILP instance generation.

Early approaches to MILP instance generation relied on domain knowledge or heuristics, designing problems with specific mathematical formulations (Rejowski Jr & Pinto, 2004; Morales-España et al., 2013; Moretti et al., 2021) or sampling instances from statistical encodings (Smith-Miles & Bowly, 2015; Bowly et al., 2020). While effective, these methods depended heavily on expert-defined templates, limiting their utility for downstream tasks such as learning-based solvers or solver tuning (Li et al., 2024). More recently, research has shifted toward learning-based paradigms that generate instances from specific problem classes, including methods for restructuring MILP formulations (Yang et al., 2024; Liu et al., 2024b), generating partial structures with Variational Autoencoders (Geng et al., 2023; Guo et al., 2024), and reconstructing constraints with diffusion models (Zhang et al., 2024).

Despite their innovation, these methods face several limitations: they require retraining separate models for each problem class, which is computationally expensive and time-consuming, and they offer limited control over the scale and difficulty of the generated instances.

To address these challenges, we propose MILP-Retrieval, a novel framework for targeted MILP instance generation that retrieves and tunes formulation code rather than reconstructing instance structures from scratch. Our method offers several advantages over prior approaches: (1) it significantly reduces the time and computational cost of instance generation; (2) it provides fine-grained control over the scale and complexity of generated instances by modifying parameters within the formulation code; and (3) it ensures that each generated instance comes with a corresponding mathematical formulation, enhancing transparency and explainability.

Our approach begins by building a large and diverse MILP library. Each

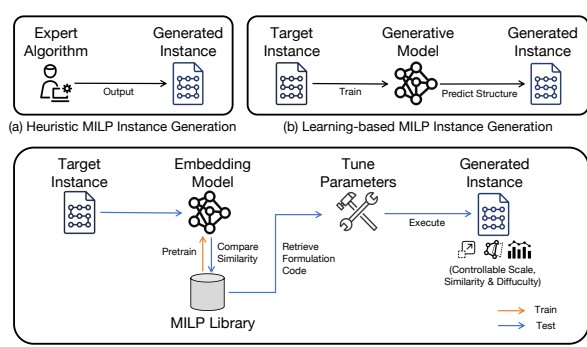

Figure 1: In MILP instance generation, (a) heuristic algorithms are used to create problem instances; (b) recent approaches train a separate model for each problem class to reconstruct problem structures; (c) our method retrieves & tunes MILP formulation code, and uses it to directly generates problem instances.

entry in this library contains a problem instance, a textual description, and the corresponding formulation code. We then pretrain an MILP embedding model using this library. Using the output of this embedding model, we introduce a novel similarity metric for MILP instances, which we refer to as *embedding metric*. Unlike conventional structural similarity metrics (Geng et al., 2023; Guo et al., 2024), our *embedding metric* capture semantic-level similarities by comparing instance embeddings. This allows for more accurate similarity measurement across instances of the same class but with different scales. MILP-Retrieval then retrieves formulation code from the MILP library based on the target instance and tunes the parameters to control the scale and difficulty of the generated instance.

We conducted extensive experiments to evaluate the generalization and robustness of our approach. The evaluation was performed on two types of datasets: (i) 50 MILP problem classes that were excluded from the training and retrieval library, and (ii) over 300 instances from the real-world benchmark MIPLIB (Gleixner et al., 2021). First, we show that our proposed *embedding metric* significantly outperforms existing metrics. We then compare the similarity between generated and target instances using both metrics. Additionally, we demonstrate the controllability of MILP-Retrieval in generating instances with varying scales and solving complexity, and we highlight how these instances can improve the performance of learning-based MILP solvers. The code and data of the paper are provided at `https://anonymous.4open.science/r/MILP-Retrieval-D830/`.

**The main contributions of the paper are as follows.**

1. We introduce a novel similarity metric for MILP instances that accurately measures the similarity between problems of the same class but different scales, addressing limitations of previous metrics.

2. We propose MILP-Retrieval, a new framework for instance generation that retrieves and tunes formulation code based on the embedding similarity metric, enabling the generation of instances highly similar to given target instances.

3. We demonstrate the practical potential of MILP-Retrieval in downstream applications, including generating instances with varying scales and difficulties and enhancing learning-based MILP solvers.

## 2 PRELIMINARY

### 2.1 MILP PROBLEM AND ITS DATA REPRESENTATIONS

The standard formulation of a Mixed-Integer Linear Programming (MILP) problem is given by:

$$\min_{x \in \mathbb{R}^n} \quad c^T x,$$
$$\text{subject to} \quad Ax \leq b,$$
$$l \leq x \leq u,$$
$$x_i \in \mathbb{Z}, \quad i \in \mathbb{I}. \tag{1}$$

In this formulation, the coefficient matrix $A \in \mathbb{R}^{m \times n}$ represents the constraints structure, $b \in \mathbb{R}^m$ denotes the constraints' right-hand side vector, and $c \in \mathbb{R}^n$ is the objective coefficient. Variables are bounded within lower $l \in (\mathbb{R} \cup \{-\infty\})^n$ and upper $u \in (\mathbb{R} \cup \{+\infty\})^n$ limits. The set $\mathbb{I} \subseteq \{1, 2, \ldots, n\}$ identifies variables constrained to integer values. We additionally utilize several alternative MILP data representations, as described below: (Figure 2 illustrates the relationships among the different forms of MILP data. For examples of these data forms, see Appendix B.3.)

**Bipartite Graph Representation**
A bipartite graph representation provides a lossless encoding of MILP problems (Gasse et al., 2019). Here, variables $\mathcal{V} = \{v_1, v_2, \ldots, v_n\}$ and constraints $\mathcal{C} = \{c_1, c_2, \ldots, c_m\}$ are represented as distinct node sets. An edge $e_{ij} = (v_i, c_j) \in \mathcal{E}$ is present if the variable $v_i$ is part of constraint $c_j$. This forms a bipartite graph $\mathcal{G} = (\mathcal{V}, \mathcal{C}, \mathcal{E})$ capturing the structural relationships between variables and constraints. Additional details regarding graph features are provided in Appendix B.1.

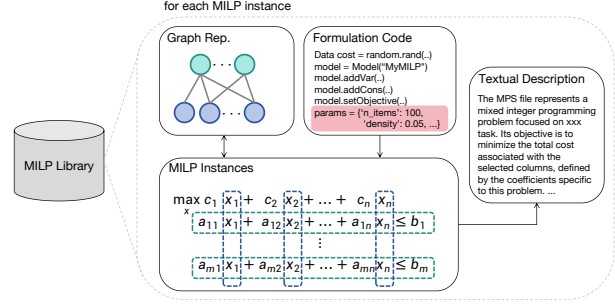

Figure 2: Relationships among different forms of MILP data.

**Formulation Code**  Formulation code represents MILP problems in a generative manner, implemented using the PySCIPOpt library (Bolusani et al., 2024). Each formulation code characterizes a distinct MILP problem class, encapsulating the procedural logic required to generate instances. As illustrated in Figure 2 (highlighted in red), the *parameter* section of the formulation code can be tuned to control various features of the generated instances, such as their size and complexity.

**Textual Description**  Textual descriptions offer natural language representations of MILP problems generated via methodologies from (Li et al., 2025). Initially, construction code is processed by a Large Language Model (LLM) to extract essential characteristics, including formulation methods and relevant topics. Statistical data of individual MILP instances are integrated to produce comprehensive descriptions combining general problem formulations and specific instance statistics.

## 2.2 MILP INSTANCE GENERATION

Prior learning-based approaches for MILP instance generation (Geng et al., 2023; Yang et al., 2024; Guo et al., 2024; Zhang et al., 2024) typically adopt a class-specific paradigm. Specifically, given a training set $P = \{p_1, p_2, \ldots, p_n\}$ belonging to a single problem class, a model is trained and subsequently used to reconstruct instances from a testing set $Q = \{q_1, q_2, \ldots, q_m\}$. The generated instances form the set $Q' = \{q'_1, q'_2, \ldots, q'_m\}$, and the primary goal is to minimize distributional divergence between $Q$ and $Q'$. For instance, previous work (Geng et al., 2023) employed Jensen-Shannon divergence (Lin, 1991) to quantify structural similarity between original and generated instances.

In this paper, we leverage MILP formulation code as backbone for targeted MILP instance generation. Under this new paradigm, a single unified model is trained on MILP problems and associated data across multiple classes, rather than being restricted to a single class. For a testing set of MILP instances $Q = \{q_1, q_2, \ldots, q_n\}$, the framework outputs a piece of MILP formulation code $c$. Executing $c$ directly produces the instance set $Q' = \{q'_1, q'_2, \ldots, q'_m\}$. The objective remains the same: to minimize the divergence between the distributions of $Q$ and $Q'$.

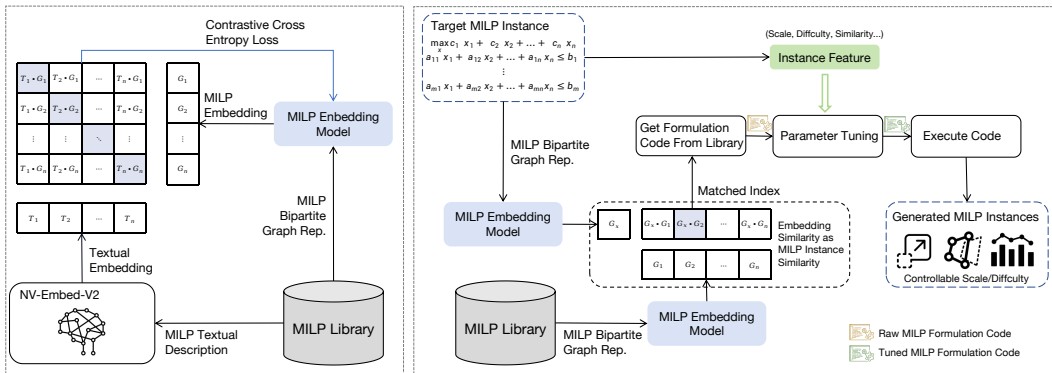

(a) Train: MILP-Language Alignment      (b) Test: Formulation Code Retrieval & Tuning

Figure 3: Our proposed framework, MILP-Retrieval, begins by constructing a comprehensive MILP library. Leveraging this library, we train an MILP embedding model following a contrastive learning paradigm. Using embeddings derived from this model, we introduce a novel similarity metric to retrieve formulation codes that best match the target instances. Subsequently, we tune the parameters within the formulation codes to control the size or difficulty of the problem. Finally, the tuned formulation codes are executed to generate the desired MILP problem instances.

## 3 METHODOLOGY

As illustrated in Figure 3, we first construct a MILP library containing diverse modalities, including MILP instances, formulation codes, bipartite graph representations, and textual descriptions (Appendix B.2). Leveraging this library, we pretrain an MILP embedding model, enabling us to map MILP instances into a unified embedding space (Section 3.1). Utilizing the pretrained embedding model, we propose a novel similarity metric designed to quantify the similarity between MILP instances, which also serves to retrieve the most relevant formulation code from the MILP library (Section 3.2). Once the appropriate formulation code is retrieved, we can further tune its parameters, enabling us to produce instances with varying scales and computational difficulties (Section 3.3).

### 3.1 PRETRAINING MILP EMBEDDING MODEL

In this subsection, we describe the pretraining process of our MILP embedding model, including the learning scheme, model architecture, training data setup, and preliminary results.

**Contrastive Learning Scheme** Using the MILP library, we train a powerful MILP embedding model capable of capturing both structural and semantic information. Specifically, we adopt a contrastive training framework inspired by CLIP (Radford et al., 2021), aligning the bipartite graph representation of MILP instances with their corresponding textual descriptions. This alignment enables the model to learn a shared embedding space that effectively captures semantic relationships between different representations of MILP problems. Our goal is to train an MILP embedding model $f_\theta : \mathcal{P} \to \mathbb{R}^d$, where $\mathcal{P}$ is the space of MILP problems. For the textual embedding component $g_\theta : \mathcal{T} \to \mathbb{R}^d$, we utilize the state-of-the-art text embedding model NV-Embed-V2 (Lee et al., 2025), freezing its weights during training. The training process employs a symmetric cross-entropy loss (Zhang & Sabuncu, 2018) designed to encourage higher similarity for correct (graph, text) pairs compared to all incorrect pairings.

**Model Architecture** Our MILP embedding model consists of two major components: (1) a bipartite Graph Neural Network (GNN) that captures the relational structure between constraints and variables, and (2) a Transformer-based self-attention module that further updates the learned representations. We represent each MILP instance as a bipartite graph $(\mathcal{V}, \mathcal{C}, \mathcal{E})$, where $\mathcal{V}$ denotes nodes corresponding to variables, $\mathcal{C}$ denotes nodes representing constraints, and $\mathcal{E}$ consists of edges connecting variables to the constraints in which they appear. To embed the nodes and edges into a shared

latent space of dimension emb_size, we employ three separate Multi-Layer Perceptrons (MLPs) for variables, constraints and edges:

$$\mathbf{x}_{u_i}^{(0)} = \text{MLP}_c(c_i), \mathbf{x}_{v_i}^{(0)} = \text{MLP}_v(v_i), \mathbf{x}_{e_{ij}} = \text{MLP}_e(e_{ij}), \tag{2}$$

where $v_i \in \mathcal{V}$, $c_i \in \mathcal{C}$, and $e_{ij} \in \mathcal{E}$ represent the raw input features, $\mathbf{x}_{u_i}^{(k)}, \mathbf{x}_{v_i}^{(k)}$ are constraint and variable embeddings at GNN layer $k$. For message passing, we utilize a Graph Convolution Module (Kipf & Welling, 2017) as the update function, the updates are performed as follows:

$$\mathbf{x}_u^{(k+1)} = \mathbf{x}_u^{(k)} + \text{BipartiteConv}\left(\mathbf{x}_v^{(k)}, \mathbf{x}_{e_{uv}}\right), \tag{3}$$

$$\mathbf{x}_v^{(k+1)} = \mathbf{x}_v^{(k)} + \text{BipartiteConv}\left(\mathbf{x}_u^{(k+1)}, \mathbf{x}_{e_{uv}}\right). \tag{4}$$

After the bipartite GNN layers, we sample $k$ (specified by hyperparameter) node embeddings randomly. Together with the mean embeddings of all variable nodes $\overline{\mathbf{x}}_v$, constraint nodes $\overline{\mathbf{x}}_u$, and the summary node $\mathbf{x}_s$, we form a set of embeddings: $\{\mathbf{x}_1, \mathbf{x}_2, \ldots, \mathbf{x}_k, \overline{\mathbf{x}}_v, \overline{\mathbf{x}}_u, \mathbf{x}_s\}$. These embeddings are then fed into Transformer encoder layers. The output of the Transformer encoder module produces a contextualized set of embeddings. We apply a final pooling operation to obtain a fixed-size embedding vector $\mathbf{z} \in \mathbb{R}^d$.

## 3.2 FORMULATION CODE RETRIEVAL

The pretrained MILP embedding model forms the backbone of a novel similarity metric, which we term the *embedding metric*. In contrast to traditional MILP similarity metrics, such as those based on the Jensen-Shannon (JS) divergence between hand-crafted statistical indicators (Geng et al., 2023; Guo et al., 2024) (referred to here as the *stat metric*, with details provided in Appendix C.2), our *embedding metric* overcomes limitations related to manual feature selection and ineffective pairwise comparisons.

Inspired by the Fréchet Inception Distance (FID) (Heusel et al., 2017; Salimans et al., 2016), a metric used in the image generation domain to evaluate the quality of generated images which employs Inception-V3 (Szegedy et al., 2016), we proposed MILP *embedding metric*. We uses the trained MILP embedding model to compute the cosine similarity between normalized embedding vectors. Formally, let $P$ and $Q$ represent two groups of MILP instances whose similarity is to be evaluated, and $f_\theta$ denote the MILP embedding model. The *embedding metric* calculation is as follows:

$$\forall p \in P, q \in Q, x_p = \frac{f_\theta(p)}{||f_\theta(p)||}, x_q = \frac{f_\theta(q)}{||f_\theta(q)||},$$
$$\text{EmbeddingMetric}(p, q) = x_p x_q^T, \tag{5}$$
$$\text{EmbeddingMetric}(P, Q) = \frac{1}{|P||Q|} \sum_{p \in P} \sum_{q \in Q} \text{EmbeddingMetric}(p, q).$$

This metric offers a major advantage over previous approaches: it enables accurate, scale-invariant similarity assessments between instances of varying sizes but belonging to the same problem class. This robustness arises from the way the embedding model is trained—instances within a problem class share similar textual descriptions, allowing the model to learn consistent cross-scale representations.

Using the pre-built MILP library as well as proposed *embedding metric*, we propose **MILP-Retrieval**, a simple yet efficient framework for MILP instance generation via formulation code retrieval. Given a group of target MILP instance $Q = \{q_1, q_2, \ldots, q_n\}$, our method retrieves the most relevant code $c_k$ from MILP library $\{(p_i, c_i)\}_{i=1}^N$, where $p_i$ represents the $i$-th instance and $c_i$ represents the corresponding code for generating that instance. The retrieval process identifies $c_k$ as:

$$k = \text{argmax}_k \sum_{i=1}^n \text{EmbeddingMetric}(q_i, p_k). \tag{6}$$

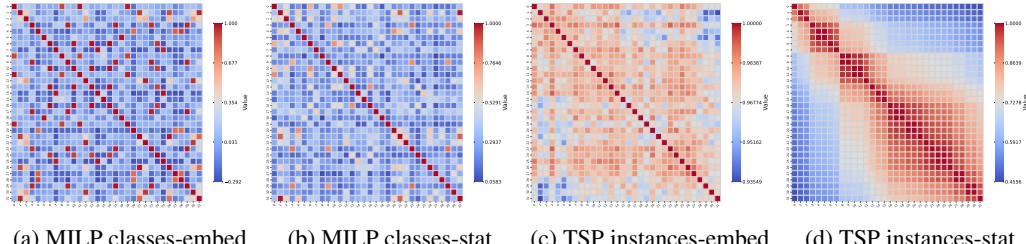

(a) MILP classes-embed     (b) MILP classes-stat     (c) TSP instances-embed     (d) TSP instances-stat

Figure 4: Comparison of similarity matrix between the *embedding metric* and the *stat metric*.

Executing $c_k$ generates new MILP instances $\{q'_1, q'_2, \ldots, q'_m\}$, effectively approximating the structural and semantic characteristics of the target instance.

### 3.3 FORMULATION CODE TUNING

Although retrieval provides formulation codes that generate semantically similar instances, additional tuning of the formulation code can further control the size and difficulty of the generated problems while preserving semantic similarity. Here, we introduce two approaches for formulation code tuning.

**Diverse Tuning** The goal of this approach is to generate problem instances that are as diverse as possible in terms of size and solving difficulty. Specifically, we randomize the parameters within the retrieved formulation codes to create multiple code variants, thereby enriching the diversity of generated instances. To automate this tuning process, modifications are restricted to numeric and interval-type parameters in the formulation code. The resulting codes are then validated and filtered to ensure the feasibility of the generated instances.

**Targeted Tuning** The objective of this approach is to achieve fine-grained control over the solving difficulty of generated problems. We treat the MILP formulation code as a black-box function: the input is the parameter configuration, and the output is the solving time of the generated MILP instance. Bayesian optimization is then employed to tune this black box. The first application of Targeted Tuning is to generate maximally difficult problems, where solving time is directly used as the optimization objective to be maximized. The second application is to generate problems with difficulty levels as close as possible to a specified target, where the optimization objective becomes the difference between the actual solving time and the target solving time, which is minimized. This tuning process is also fully automated by parsing the tunable parameters from the formulation code and configuring the parameter space for the Bayesian optimizer.

Together, these two tuning strategies enhance the practicality of MILP-Retrieval and provide greater control over the size and difficulty of generated problems. Further technical details of the formulation code tuning procedure are provided in Appendix B.5.

### 4 EXPERIMENTS

We firstly evaluate the proposed *embedding metric* by comparing it against existing *stat metric*, to demonstrate its superior accuracy. Second, we assess the quality of MILP instances generated by MILP-Retrieval. The generated instances are evaluated using similarity metrics and compared against instances produced by several baselines. Additionally, we evaluate the performance of MILP-Retrieval on downstream tasks. These tasks include improving the performance of learning-based MILP solvers and generating MILP instances with varying scales and difficulty levels.

### 4.1 EXPERIMENTAL SETUP

**Datasets** We conduct experiments on two datasets to ensure a fair and comprehensive evaluation: (1) the *Evolve/Test* dataset, containing 50 distinct problem classes, and (2) the widely-used MIPLIB benchmark (Gleixner et al., 2021). MILP-Retrieval utilizes *Evolve/Train* as the retrieval library. For

each problem class in *Evolve/Test*, we generate 20 instances that serve both as the training set for learning-based baselines and as the target instances for MILP-Retrieval. For MIPLIB, we manually define problem class partitions to support evaluation. Further details on the datasets are provided in Appendix C.3.

**Metrics** We employed multiple metrics to comprehensively evaluate our proposed approach. Specifically, we evaluate the similarity between generated instances and target instances using both the proposed *embedding metric* and traditional *stat metric*. Since MILP-Retrieval can tune formulation code to generate instances at different scales and difficulties, we utilize Gurobi (Gurobi Optimization, LLC, 2024) to solve both the target instances and the generated instances, reporting the solving time. Details on the calculation of the *stat metric* are provided in Appendix C.2.

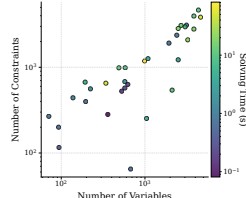

Figure 5: Problem size and solving time of 32 TSP instances.

**Baselines** We compare MILP-Retrieval against a diverse set of baselines. For heuristic generation method, we compare against *Bowly* (Bowly et al., 2020). For learning-based methods, we evaluate against the state-of-the-art open-source method *ACM-MILP* (Guo et al., 2024), which adopts a Variational Autoencoder (VAE) framework. We further implement two LLM-based baselines *GPT-4o* (Hurst et al., 2024) and *Finetuned LLaMA3-8b* (Dubey et al., 2024), which directly generate MILP formulation codes from textual descriptions, serving as baselines that generate instances via MILP formulation code. Implementation details for baselines are provided in Appendix C.1.

## 4.2 MILP SIMILARITY METRIC COMPARISON

To illustrate the effectiveness of proposed metric, we conducted two sets of comparative experiments between the *embedding metric* and *stat metric*.

In the first experiment, we evaluated the similarity among the first 32 MILP problem classes in the *Evolve/Train* dataset. The resulting similarity matrices obtained using the embedding metric and the stat metric are shown in Figure 4a and 4b, respectively. As illustrated, the *embedding metric* similarity matrix reveals many high-similarity MILP class pairs that are not captured by the stat metric. This observation aligns with the design of MILP-Evolve, which constructs problem classes through evolutionary mechanisms, resulting in semantically related instances.

In the second experiment, we generated 32 TSP instances of varying sizes. We firstly visualize their size and solving time (computed by Gurobi) in Figure 5. The similarity matrices derived using the *embedding metric* and the *stat metric* are presented in Figure 4c and Figure 4d. The instances are ordered by problem size in the matrices. Our results demonstrate that embedding metric generalizes effectively to unseen instances, providing robust similarity measurements for unseen MILP instances.

## 4.3 RESULTS ON TARGETED MILP INSTANCE GENERATION

We report the similarity between the generated instances and the target instances using both the *embedding metric* and the *stat metric* in Table 1 and Table 2. For LLM-based methods, evaluation is limited to the *Evolve/Test* dataset, as generating MILP formulation codes from textual descriptions is currently only feasible in this setting. Due to the fact that learning-based methods (e.g., ACM-MILP (Guo et al., 2024)) require training a separate model for each problem class, we could not evaluate them across all MILP classes in *Evolve/Test* and MIPLIB. Instead, we selected four problem classes from *Evolve/Test*: FCNF, TSP, GA, VRP, as well as three widely studied problem classes from MIPLIB: Nursesched, CVS, and IIS.

From the results, we observe that MILP-Retrieval significantly outperforms baselines under the *embedding metric*, but performs less competitively under the *stat metric* compared to learning-based methods. This is expected, as our framework is designed to generate problem instances that are semantically similar to the target instances, without necessarily matching their statistical characteristics. We further discuss the experimental results in Appendix E.

Table 1: Comparison between generated instances and target instances on the *embedding metric*.

| | Method | MILP-Retrieval | Bowly | ACM-MILP | GPT-4o | Finetuned LLaMA 3-8b |
|---|---|---|---|---|---|---|
| Evolve/Test | FCNF | **0.705 ± 0.174** | -0.079 ± 0.088 | 0.419 ± 0.143 | infeasible | 0.076 ± 0.094 |
| | TSP | **0.920 ± 0.050** | 0.041 ± 0.039 | infeasible | 0.304 ± 0.073 | 0.399 ± 0.011 |
| | GA | **0.734 ± 0.078** | 0.167 ± 0.087 | 0.015 ± 0.031 | infeasible | 0.233 ± 0.027 |
| | VRP | **0.960 ± 0.015** | 0.005 ± 0.055 | infeasible | 0.347 ± 0.053 | infeasible |
| MIPLIB | Nursesched | **0.883 ± 0.085** | 0.071 ± 0.042 | -0.056 ± 0.108 | - | - |
| | CVS | **0.814 ± 0.078** | -0.105 ± 0.080 | 0.030 ± 0.106 | - | - |
| | IIS | **0.829 ± 0.046** | -0.119 ± 0.030 | -0.210 ± 0.024 | - | - |

Table 2: Comparison between generated instances and target instances on the *stat metric*.

| | Method | MILP-Retrieval | Bowly | ACM-MILP | GPT-4o | Finetuned LLaMA 3-8b |
|---|---|---|---|---|---|---|
| Evolve/Test | FCNF | 0.611 ± 0.006 | 0.530 ± 0.019 | **0.795 ± 0.018** | infeasible | 0.568 ± 0.022 |
| | TSP | 0.367 ± 0.129 | 0.665 ± 0.043 | infeasible | **0.840 ± 0.059** | 0.469 ± 0.034 |
| | GA | 0.436 ± 0.014 | 0.479 ± 0.016 | **0.703 ± 0.003** | infeasible | 0.311 ± 0.002 |
| | VRP | 0.377 ± 0.024 | **0.672 ± 0.021** | infeasible | 0.599 ± 0.002 | infeasible |
| MIPLIB | Nursesched | 0.231 ± 0.076 | 0.313 ± 0.046 | **0.655 ± 0.032** | - | - |
| | CVS | 0.430 ± 0.121 | 0.417 ± 0.032 | **0.717 ± 0.019** | - | - |
| | IIS | 0.234 ± 0.003 | 0.365 ± 0.004 | **0.878 ± 0.059** | - | - |

Table 3: The performance of Neural Diving on test set of 4 classes of problems. We use each method to generate 40 instances and add them to the training set, we mark the best performance in bold.

| | Raw | MILP-Retrieval | ACM-MILP | GPT-4o | Finetuned LLaMA 3-8b |
|---|---|---|---|---|---|
| FCNF | 1604.77 ± 311.27 | **1117.14 ± 187.68** | 1520.08 ± 200.01 | - | 1228.64 ± 399.68 |
| TSP | 944.85 ± 98.45 | 893.47 ± 83.86 | - | **891.53 ± 83.79** | 924.40 ± 86.83 |
| GA | -49768.53 ± 53.77 | **-49991.25 ± 4.92** | infeasible | - | -49293 ± 15.23 |
| VRP | 911.20 ± 91.02 | **774.91 ± 52.64** | - | 827.66 ± 40.98 | - |

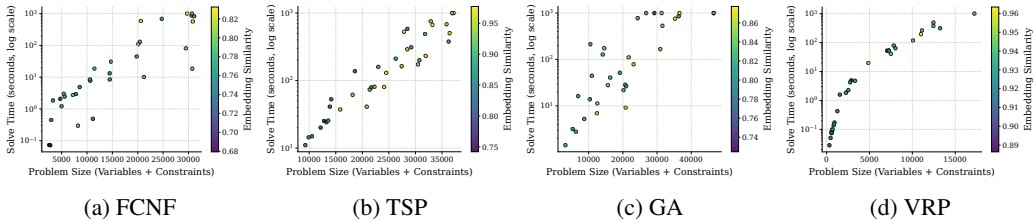

| (a) FCNF | (b) TSP | (c) GA | (d) VRP |
|---|---|---|---|

Figure 6: Visualization of generated scalable instances through formulation code *diverse tuning*.

## 4.4 RESULTS ON DOWNSTREAM TASKS

**Enhancing Learning-based Solver** We evaluated our method on multiple ML-based solvers, including Neural Diving, Predict-and-Search, and Learn-to-Branch (Nair et al., 2020; Han et al., 2023; Gasse et al., 2019). Experiments are also conducted on four problem classes: FCNF, TSP, GA, and VRP. MILP-Retrieval and baseline methods are then used to generate varying numbers of supplementary training instances, which are added to the original training set of Neural Diving. The enhanced models are evaluated on the test set. Experimental results of Neural Diving summarized in Table 3 report the objective value for each experiment. Our findings demonstrate that MILP-Retrieval achieves comparable or superior performance to baseline approaches in boosting solver performance. Details of this experiment and additional results can be found in Appendix D.4.

**Controllable MILP Instance Generation by *Diverse Tuning*** We apply *Diverse Tuning* to generate instances that vary widely in both scale and solving difficulty. Experiments are conducted on the same four problem classes, where 32 instances are generated per class under different parameter settings. Figure 6 illustrates the distributions of instance sizes and solving times (measured by Gurobi (Gurobi Optimization, LLC, 2024), with a maximum time limit of 1000s) for each class. The results demonstrate that formulation code tuning effectively enables the generation of MILP instances from the same class that differ substantially in scale and difficulty.

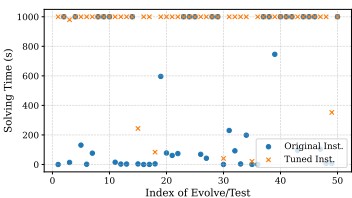

Figure 7: *Targeted Tuning* for maximizing difficulty.

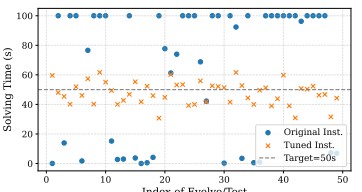

Figure 8: *Targeted Tuning* for matching specified difficulty.

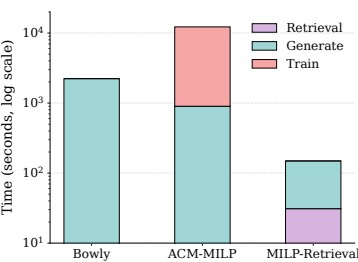

Figure 9: Computational efficiency of Different Methods.

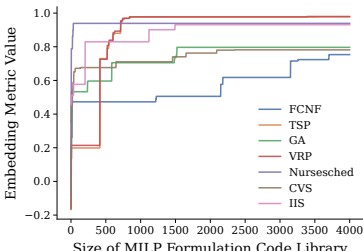

Figure 10: Ablation experiment on formulation code library size.

**Controllable MILP Instance Generation by *Targeted Tuning*** We further showcase the use of *Targeted Tuning* to precisely control the solving difficulty of MILP instances. The experiments are performed on the 50 MILP classes from Evolve/Test. In the first case, the goal is to maximize difficulty. We set both the target difficulty and the solving time cutoff to 1000s. As shown in Figure 7, Targeted Tuning successfully adjusts 88% of the 50 classes to reach the desired 1000s solving difficulty. In the second case, the goal is to match specified difficulty level as closely as possible. Using a target solving time of 50s as example (results shown in Figure 8), the generated instances achieve an average deviation of only 12.8% from the target, a substantial improvement over the original instances without tuning. These results verify the effectiveness of formulation code tuning within MILP-Retrieval. Additional experimental details and results are provided in Appendix B.5.

### 4.5 EXTENSIVE STUDIES

**Computational Efficiency** We use FCNF as a case study to demonstrate the significant improvement in computational efficiency achieved by MILP-Retrieval. We measure the time required to train the model and generate 1,000 instances, with the results shown in Figure 9. It is worth-noting that training the MILP embedding model took approximately 40 hours. We exclude this from the comparison, as the embedding model is designed to be generalized across different problem classes.

**Ablation Study** To evaluate the influence of formulation code library size, we limit the size of the retrieval library and observe how similarity (*embedding metric*) between target instances and generated instances changes with reduced library size. The results are reported in Figure 10, show that our current MILP library is sufficiently large to support robust instance generation.

### 5 CONCLUSION

In this paper, we propose MILP-Retrieval, a framework for targeted MILP instance generation via formulation code retrieval and parameter tuning. It provides a generalizable solution that efficiently generates problem instances of varying difficulty and scale, thereby improving the performance of learning-based solvers. While its effectiveness depends on the size of the formulation code library, we also explore LLM-based methods to directly generate formulation code from textual descriptions as a baseline. Advancing LLM-based approaches for fine-grained and controllable generation remains a promising direction for future research.

ETHICS STATEMENT

The methods proposed in this paper aim to retrieve and tune MILP formulation code for MILP instance generation, which is related to the broader field of neural combinatorial optimization. To our best knowledge, no ethical issues or harmful insights of this work need to be otherwise stated.

REPRODUCIBILITY STATEMENT

The datasets used and the baseline implementation are described in Appendix C. The detailed hypermeters and implementation of the models for training and testing are provided in Appendix B. Source code and datasets can be accessed at `https://anonymous.4open.science/r/MILP-Retrieval-D830/`.

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

TABLE OF CONTENTS FOR APPENDIX

## A    RELATED WORKS

**Machine Learning on MILP**    Machine learning methods have demonstrated superior performance over traditional algorithms in solving various combinatorial optimization problems due to their ability to capture the characteristics of similar problems. These approaches can be broadly classified into two categories. The first category involves integrating learning-based modules into traditional solvers by replacing or augmenting key components, such as branching (Gasse et al., 2019; Gupta et al., 2020; 2022), cut selection (Tang et al., 2020; Wang et al., 2023), and presolve (Kuang et al., 2023; Liu et al., 2024a). The second category focuses on improving the solution search process itself. Techniques such as predict-and-optimize (Han et al., 2023; Ye et al., 2023; 2024) and large neighborhood search (Sonnerat et al., 2021; Huang et al., 2023) utilize predictive models to guide the solver toward promising regions of the solution space, thereby enhancing efficiency and solution quality. A key challenge in both categories is the availability of sufficient MILP data for training these models. This challenge highlights the critical need for generating diverse and high-quality MILP instances.

**MILP Instance Generation**    The field of MILP instance generation has traditionally relied on heuristic methods to create problem instances tailored to specific types or statistical characteristics (Smith-Miles & Bowly, 2015; Bowly et al., 2020). While effective in controlled scenarios, these methods often lack the flexibility to address broader applications or more diverse instance distributions. Learning-based MILP generation methods use model to learn the distribution of the problems and reconstruct them. For example, some methods focus on restructuring the problem's underlying structure (Liu et al., 2024b; Yang et al., 2024), while others utilize paradigms like VAE or diffusion models to reconstruct problem constraints (Geng et al., 2023; Wang et al., 2024; Guo et al., 2024; Zhang et al., 2024). Recent work (Li et al., 2025) proposes a novel approach for generating diverse MILP problems. Our approach MILP-Retrieval, along with the concept of MILP *embedding metric*, offers a novel perspective on MILP instance generation.

## B    IMPLEMENTATION DETAILS OF MILP-RETRIEVAL

### B.1    DETAILS OF BIPARTITE GRAPH FEATURES

To encode an MILP instance as a corresponding bipartite graph, we incorporate information about both variables and constraints into the node features of the graph representation. The specific node features used in our encoding are detailed in Table 4.

Additionally, the bipartite graph features include solution-related information about the MILP instance. To obtain this data, we solve each problem instance using Gurobi (Gurobi Optimization, LLC, 2024), with a computation time limit of 50 seconds per instance. This ensures a standardized and practical approach to extracting solution-based features while maintaining computational efficiency.

### B.2    DETAILS OF MILP LIBRARIES *Evolve/Train* AND *Evolve/Test*

We construct the MILP libraries following the method proposed in MILP-Evolve (Li et al., 2025), which leverages LLMs to evolve MILP formulation code and generate diversified MILP instances. This approach guarantees that all generated instances are feasible. Based on this method, we build two separate libraries—*Evolve/Train* and *Evolve/Test*—for training and testing purposes, respectively.

The *Evolve/Train* library is evolved from eight seed classes (IS, SC, CA, CFL, KS, GIS, NF, and SAT), resulting in 4,000 formulation codes and 59,033 corresponding MILP instances, graphs, and textual descriptions. This library is used both to train the MILP embedding model and as the retrieval corpus for MILP-Retrieval. In contrast, the *Evolve/Test* library is evolved from four disjoint seed classes (FCNF, TSP, GA, and VRP), yielding 50 formulation codes and 672 corresponding MILP instances, graphs, and textual descriptions. The seed classes of *Evolve/Train* and *Evolve/Test* are completely disjoint.

Table 4: Node type features and descriptions for Variables and Constraints.

| Node Type | Feature | Description |
|---|---|---|
| **Vars** | norm coef | Objective coefficient, normalized by objective norm |
| | type | Var type (binary, integer, impl. integer, continuous), one-hot |
| | has lb | Lower bound indicator |
| | has ub | Upper bound indicator |
| | solval | Solution value |
| | solfrac | Solution value fractionality |
| | sol_is_at_lb | Solution value equals lower bound |
| | sol_is_at_ub | Solution value equals upper bound |
| | basestat | Simplex basis status (lower, basic, upper, zero), one-hot |
| **Cons** | rank | Rank of a row |
| | norm_nnzrs | Fraction of nonzero entries |
| | bias | Unshifted side normalized by row norm |
| | row_is_at_lhs | Row value equals left hand side |
| | row_is_at_rhs | Row value equals right hand side |
| | dualsol | Dual LP solution of a row, normalized by row and objective norm |
| | norm_intcols | Fraction of integral columns in the row |

Each MILP class generates 20 instances, which are subsequently filtered by solving with a time limit of 50 seconds; instances without feasible solutions are removed. After filtering, the final datasets consist of 59,033 instances in *Evolve/Train* and 672 instances in *Evolve/Test*.

The evolution process was carried out using GPT-4o-mini as the LLM. Starting from the seed classes, constructing both libraries required approximately four weeks and incurred a total cost of around $50. Further details on the class generation procedure can be found in (Li et al., 2025). The sources of the seed classes are summarized in Table 5, and the distribution of variables and constraints in *Evolve/Train* is visualized in Figure 11.

Table 5: 8 Seed Classes for *Evolve/Train* and 4 Seed Classes for Evolve/Test.

| Dataset | Abbreviation | Full Name | Reference |
|---|---|---|---|
| **Evolve/Train** | IS | Maximum Independent Set | (Bergman et al., 2016) |
| | SC | Set Cover | (Balas & Ho, 1980) |
| | CA | Combinatorial Auction | (Leyton-Brown et al., 2000) |
| | CFL | Capacitated Facility Location | (Cornuéjols et al., 1991) |
| | Knapsack | Multiple Knapsack | (Pisinger, 1999) |
| | GIS | Generalized Independent Set | (Colombi et al., 2017) |
| | NF | Multicommodity Network Flow | (Hewitt et al., 2010) |
| | SAT | Max Satisfiability | (Béjar et al., 2009) |
| **Evolve/Test** | FCNF | Fixed-Charge Network Flow | (Kim & Pardalos, 1999) |
| | TSP | Traveling Salesman Problem | (Matai et al., 2010) |
| | GA | Generalized Assignment | (Cattrysse & Van Wassenhove, 1992) |
| | VRP | Vehicle Routing Problem | (Braekers et al., 2016) |

### B.3 SAMPLES OF DIFFERENT FORMS OF MILP DATA

Here we provide a sample of code and textual description in MILP Data, which comes from the Set Cover problem and is one of the seed classes of *Evolve/Train*. Lines 91-96 in the code correspond to the parameter part of the code, which can be used by formulation code tuning to adjust the size and difficulty of the generated instance.

> **Formulation Code**

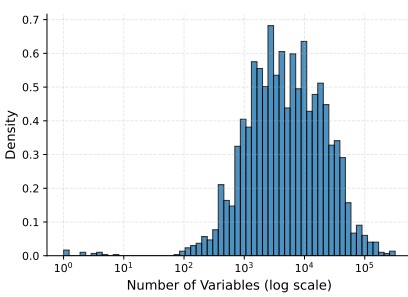

(a) Distribution of Number of Variables.

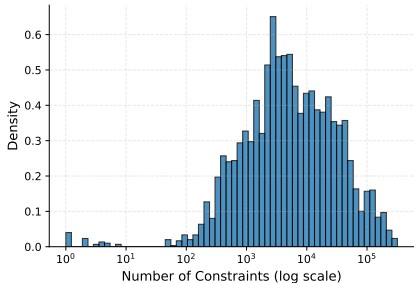

(b) Distribution of Number of Constraints.

Figure 11: Visualization of All Instances in Evolve/Train.

```
1   import random
2   import time
3   import scipy
4   import numpy as np
5   import networkx as nx
6   from pyscipopt import Model, quicksum
7
8   class SetCover:
9       def __init__(self, parameters, seed=None):
10          for key, value in parameters.items():
11              setattr(self, key, value)
12
13          self.seed = seed
14          if self.seed:
15              random.seed(seed)
16              np.random.seed(seed)
17
18      ################ Data Generation ################
19      def generate_instance(self):
20          nnzrs = int(self.n_rows * self.n_cols * self.density)
21
22          # compute number of rows per column
23          indices = np.random.choice(self.n_cols, size=nnzrs)  #
                random column indexes
24          indices[:2 * self.n_cols] = np.repeat(np.arange(self.
                n_cols), 2)  # force at leats 2 rows per col
25          _, col_nrows = np.unique(indices, return_counts=True)
26
27          # for each column, sample random rows
28          indices[:self.n_rows] = np.random.permutation(self.
                n_rows) # force at least 1 column per row
29          i = 0
30          indptr = [0]
31          for n in col_nrows:
32              # empty column, fill with random rows
33              if i >= self.n_rows:
34                  indices[i:i+n] = np.random.choice(self.n_rows,
                        size=n, replace=False)
35
36              # partially filled column, complete with random rows
                    among remaining ones
37              elif i + n > self.n_rows:
38                  remaining_rows = np.setdiff1d(np.arange(self.
                        n_rows), indices[i:self.n_rows],
                        assume_unique=True)
```

```
39                    indices[self.n_rows:i+n] = np.random.choice(
                          remaining_rows, size=i+n-self.n_rows,
                          replace=False)
40
41              i += n
42              indptr.append(i)
43
44          # objective coefficients
45          c = np.random.randint(self.max_coef, size=self.n_cols) +
                 1
46
47          # sparce CSC to sparse CSR matrix
48          A = scipy.sparse.csc_matrix(
49                  (np.ones(len(indices), dtype=int), indices,
                        indptr),
50                  shape=(self.n_rows, self.n_cols)).tocsr()
51          indices_csr = A.indices
52          indptr_csr = A.indptr
53
54          res =  {'c': c,
55                  'indptr_csr': indptr_csr,
56                  'indices_csr': indices_csr}
57
58          return res
59
60      ################ PySCIPOpt Modeling ################
61      def solve(self, instance):
62          c = instance['c']
63          indptr_csr = instance['indptr_csr']
64          indices_csr = instance['indices_csr']
65
66          model = Model("SetCover")
67          var_names = {}
68
69          # Create variables and set objective
70          for j in range(self.n_cols):
71              var_names[j] = model.addVar(vtype="B", name=f"x_{j}"
                    , obj=c[j])
72
73          # Add constraints to ensure each row is covered
74          for row in range(self.n_rows):
75              cols = indices_csr[indptr_csr[row]:indptr_csr[row +
                    1]]
76              model.addCons(quicksum(var_names[j] for j in cols)
                    >= 1, f"c_{row}")
77
78          # Set objective: Minimize total cost
79          objective_expr = quicksum(var_names[j] * c[j] for j in
                range(self.n_cols))
80
81          model.setObjective(objective_expr, "minimize")
82
83          start_time = time.time()
84          model.optimize()
85          end_time = time.time()
86
87          return model.getStatus(), end_time - start_time
88
89  if __name__ == '__main__':
90      seed = 42
91      parameters = {
92          'n_rows': 750,
93          'n_cols': 1500,
```

```
94              'density': 0.05,
95              'max_coef': 100,
96          }
97
98      set_cover_problem = SetCover(parameters, seed=seed)
99      instance = set_cover_problem.generate_instance()
100     solve_status, solve_time = set_cover_problem.solve(instance)
101
102     print(f"Solve Status: {solve_status}")
103     print(f"Solve Time: {solve_time:.2f} seconds")
```

**Textual Description**

The MPS file, named 'SetCover', represents a mixed integer programming problem focused on a Set Cover optimization task. Its objective is to minimize the total cost associated with the selected columns, defined by the coefficients specific to this problem. The formulation leverages inequalities to ensure that each of the 750 constraints guarantees that every row is covered by at least one selected column. The decision variables are binary, reflecting the choice of each column's inclusion in the cover. The file employs a structured approach for encoding the problem, facilitating efficient solving by optimization algorithms.

### B.4 DETAILS OF MILP EMBEDDING MODEL

Below we provide more details about the embedding model.

#### B.4.1 DERIVATION OF LOSS FUNCTION

Let $(\mathcal{P}_i, \mathcal{T}_i)$ for $i = 1, \ldots, N$ be a batch of $N$ matched MILP–text pairs, $f_\theta$ be the MILP embedding model, producing an MILP embedding $\mathbf{p}_i = f_\theta(\mathcal{P}_i) \in \mathbb{R}^d$, $g_\theta$ be the text encoder, producing a text embedding $\mathbf{t}_i = g_\theta(\mathcal{T}_i) \in \mathbb{R}^d$. Both $\mathbf{p}_i$ and $\mathbf{t}_i$ are typically L2-normalized to have unit length, $\|\mathbf{p}_i\|_2 = 1, \|\mathbf{t}_i\|_2 = 1$. For each MILP–text pair $(i, j)$ in the batch, we define the similarity score as the dot product: $s_{ij} = \mathbf{p}_i^\top \mathbf{t}_j$.

Our training objective is a bidirectional contrastive objective: it treats each MILP instance $\mathbf{p}_i$ as a query and tries to classify the correct text $\mathbf{t}_i$ among all texts $\{\mathbf{t}_j\}$, and symmetrically, each text $\mathbf{t}_i$ tries to classify the correct MILP instances $\mathbf{v}_i$ among all instances $\{\mathbf{v}_j\}$.

For a fixed MILP embedding $\mathbf{p}_i$, the MILP-to-text cross-entropy loss is:

$$\ell_i^{\text{(MILP-to-Text)}} = -\log\left(\frac{\exp(s_{ii})}{\sum_{j=1}^{N} \exp(s_{ij})}\right)$$

Similarly, for a fixed text embedding $\mathbf{t}_i$, the text-to-MILP cross-entropy loss is:

$$\ell_i^{\text{(Text-to-MILP)}} = -\log\left(\frac{\exp(s_{ii})}{\sum_{j=1}^{N} \exp(s_{ji})}\right)$$

To incorporate both MILP-to-text and text-to-MILP objectives, the final symmetric loss sums these two cross-entropy terms for each pair and then averages over the batch:

$$\mathcal{L} = \frac{1}{2N} \sum_{i=1}^{N} \ell_i^{\text{(MILP-to-Text)}} + \ell_i^{\text{(Text-to-MILP)}}$$

### B.4.2 PROMPT DETAILS OF NV-EMBED-V2

We use NV-Embed-v2 (Lee et al., 2025) as the text embedding model in the training paradigm (see Figure 3a) and freeze its parameters during training. NV-Embed-V2 is an instruction embedding model, and we use the following prompt as the instruction:

---

**Prompt for Text Embedding Model**

Given a linguistic description, retrieve the corresponding Mixed-Integer Linear Programming problem.

---

### B.4.3 TRAINING DETAILS

We trained the MILP embedding model on the *Evolve/Train* dataset, which contains a total of 59,033 (MILP instance, textual description) pairs. We randomly divided it into a training set and a validation set in a ratio of 9:1, using the training set as training data. To evaluate training progress, we track 4-way and 10-way retrieval accuracies on the validation set, which measure whether the model can correctly match a MILP problem to its textual description (or vice versa) among 4 or 10 candidates, respectively. Figure 12 shows the validation accuracy curves during training, demonstrating that the model effectively learns to capture the semantics of MILP problems through our proposed contrastive framework. These retrieval accuracies serve as intermediate metrics for assessing the quality of the learned MILP embeddings. The training process was completed on a single Nvidia H100 and took about 40 hours. We provide the hyperparameters used for training in Table 6.

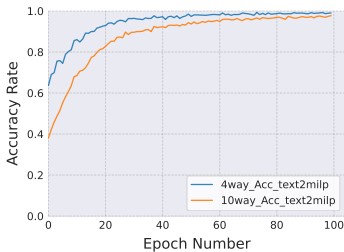 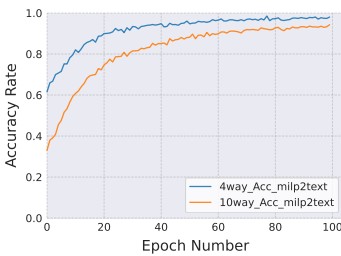

Figure 12: The Text-to-MILP and MILP-to-Text Accuracy Rate curves with respect to epoch number.

Table 6: Hyperparameter of MILP embedding model.

| Name | Value | Name | Value |
|---|---|---|---|
| Embed_Size | 64 | Num. of GCN Layers | 2 |
| Num. of Sampled Nodes | 512 | Num. of Attention Layers | 6 |
| Embedding Space | $\mathbb{R}^{4096}$ | Epoch Number | 100 |
| Learning Rate | 0.001 | Batch Size | 64 |
| Num. of Attention Heads | 8 | Optimizer | Adam |

### B.5 DETAILS OF FORMULATION CODE TUNING

This appendix expands the implementation details for the two tuning strategies introduced in Section 3.3. We first parse the formulation code using Python's `ast` to locate tunable parameters in the "parameter" block. Two parameter types are supported: (i) *value* (scalar numeric, integer or real), and (ii) *interval* (lower/upper bounds). For *Diverse Tuning* (Algorithm 1), we draw multiplicative scale factors from a log-uniform range and rewrite the code accordingly. For *Targeted Tuning* (Algorithm 2), we treat the formulation as a black-box generator and apply Bayesian Optimization (BO) over the parameter space. Each candidate parameterization yields a temporary instance that is solved (with a time limit); the result is used to accept/reject instances (diverse tuning) or to guide the BO loop (targeted tuning).

In the *Diverse Tuning* setting, we (i) sample multiplicative scales from $[0.1, 10]$ on a log scale; (ii) preserve integer parameters by rounding to the nearest valid integer; (iii) keep interval ordering by enforcing $\ell' < u'$ (with a small jitter if needed); (iv) discard infeasible instances and instances solved in less than 5 seconds as trivial; and (v) cap solving at 1000 seconds (treating timeouts as 1000). All experiments use PySCIPOpt 5.2.1 (Bolusani et al., 2024).

In the *Targeted Tuning* setting, we employ smac3 (Lindauer et al., 2022) as the Bayesian optimizer, with a maximum of 50 trials. For cases where the objective is to maximize solving time, we cap the runtime of each trial at 1000s. For cases where the goal is to match a specified solving time, we set the target time to 50s and limit each trial to 100s. Optimization traces for both cases are provided in Figures 13 and 14 as illustrative examples.

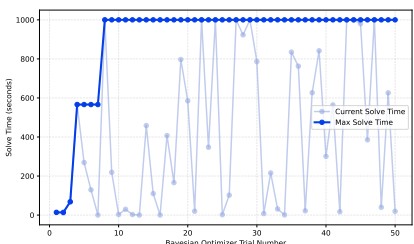

Figure 13: Example optimization trace for the case of maximizing solving time.

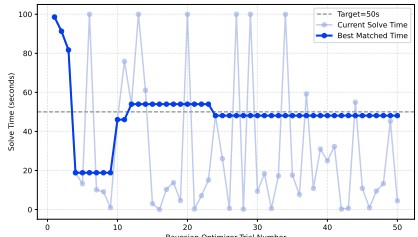

Figure 14: Example Optimization trace for the case of matching a specified target time (50s).

---

**Algorithm 1** Diverse Tuning via Randomized Parameter Perturbations

---

**Require:** Formulation code $c$; desired number of accepted instances $k$; scale range $[a, b]$ (default $[0.1, 10]$); trivial cutoff $t_{\min}$ (default 5 s); time limit $t_{\max}$ (default 1000 s)
**Ensure:** Accepted instances $\mathcal{Q} = \{(q_i, \tau_i, \mathbf{w}_i)\}_{i=1}^{k}$ with solve time $\tau_i \geq t_{\min}$
1: $\mathcal{Q} \leftarrow$,   Success $\leftarrow 0$
2: Parse tunable parameters $\mathbf{w} = (w_1, \ldots, w_n)$ from the parameter block of $c$ {$w_i$ can be a scalar or an interval $[\ell_i, u_i]$}
3: **while** Success $< k$ **do**
4:    **for** $i = 1$ **to** $n$ **do**
5:       **if** $w_i$ is a scalar value **then**
6:          Draw $s_i \sim \text{LogUniform}(a, b)$
7:          $w'_i \leftarrow \text{round\_if\_int}(w_i \cdot s_i)$
8:          $w'_i \leftarrow \text{clamp\_if\_bounded}(w'_i)$
9:       **else if** $w_i$ is an interval $[\ell_i, u_i]$ **then**
10:          Draw $s_i \sim \text{LogUniform}(a, b)$
11:          $\ell'_i \leftarrow \text{round\_if\_int}(\ell_i \cdot s_i)$,   $u'_i \leftarrow \text{round\_if\_int}(u_i \cdot s_i)$
12:          **if** $\ell'_i \geq u'_i$ **then**
13:             $(\ell'_i, u'_i) \leftarrow \text{repair}(\ell'_i, u'_i)$
14:          **end if**
15:          $w'_i \leftarrow [\ell'_i, u'_i]$
16:       **end if**
17:    **end for**
18:    $c' \leftarrow \text{SETPARAMETER}(c, \mathbf{w}')$
19:    $q' \leftarrow \text{GENERATEINSTANCE}(c')$
20:    $(\text{feasible}, \tau) \leftarrow \text{SOLVE}(q', t_{\max})$ {$\tau$ is wall-clock solve time; timeouts yield $\tau = t_{\max}$}
21:    **if** feasible **and** $\tau \geq t_{\min}$ **then**
22:       $\mathcal{Q} \leftarrow \mathcal{Q} \cup \{(q', \tau, \mathbf{w}')\}$,   Success $\leftarrow$ Success $+ 1$
23:    **end if**
24: **end while**
25: **return** $\mathcal{Q}$

---

---

**Algorithm 2** Targeted Tuning via Black-Box Optimization

---

**Require:** Formulation code $c$; evaluation budget $B$; time limit $t_{\max}$; trivial cutoff $t_{\min}$; objective obj $\in \{\text{MAXDIFFICULTY}, \text{HITTARGET}\}$; target time $T$ (only if obj $=$ HITTARGET)
**Ensure:** Best parameterization $\mathbf{x}^\star$, instance $q^\star$, and measured solve time $\tau^\star$
1: Construct search domain $\mathcal{X}$ from tunable parameters in $c$ (respect bounds and integrality)
2: **Define** EVALUATE($\mathbf{x}$):
3:    $c' \leftarrow$ SETPARAMETER$(c, \mathbf{x})$;   $q \leftarrow$ GENERATEINSTANCE$(c')$
4:    (feasible, $\tau$) $\leftarrow$ SOLVE$(q, t_{\max})$ {Timeouts return $\tau = t_{\max}$}
5:    **if** not feasible **or** $\tau < t_{\min}$ **then**
6:      $y \leftarrow \begin{cases} 0 & \text{if obj} = \text{MAXDIFFICULTY} \\ -|t_{\min} - T| & \text{if obj} = \text{HITTARGET} \end{cases}$
7:    **else**
8:      $y \leftarrow \begin{cases} \tau & \text{if obj} = \text{MAXDIFFICULTY} \\ -|\tau - T| & \text{if obj} = \text{HITTARGET} \end{cases}$
9:    **end if**
10:   **return** $(y, q, \tau)$
11: Initialize a black-box optimizer $\mathsf{O} \leftarrow$ INITIALIZEBO$(\mathcal{X})$
12: Optionally warm-start $\mathsf{O}$ with a few random evaluations of EVALUATE$(\cdot)$
13: Incumbent $\leftarrow$ None
14: **for** $t = 1$ **to** $B$ **do**
15:   $\mathbf{x}_t \leftarrow \mathsf{O}.$PROPOSE$()$
16:   $(y_t, q_t, \tau_t) \leftarrow$ EVALUATE$(\mathbf{x}_t)$
17:   $\mathsf{O}.$OBSERVE$(\mathbf{x}_t, y_t)$
18:   **if** Incumbent $=$ None **or** $y_t$ improves over Incumbent.$y$ **then**
19:     Incumbent $\leftarrow (\mathbf{x}_t, q_t, \tau_t, y_t)$
20:   **end if**
21: **end for**
22: $(\mathbf{x}^\star, q^\star, \tau^\star, \_) \leftarrow$ Incumbent
23: **return** $(\mathbf{x}^\star, q^\star, \tau^\star)$

---

## C   MORE DETAILS ON EXPERIMENTAL SETUP

### C.1   DETAILS OF BASELINES

#### C.1.1   'GPT-4O' BASELINE

Our implemented 'GPT-4o' baseline uses GPT-4o (Hurst et al., 2024) as the underlying LLM to evaluate its capability to directly generate MILP formulation code from textual description. We conduct experiments on the *Evolve/Test* dataset, which contains instances paired with textual descriptions. A few-shot prompting approach is employed to guide the LLM in generating MILP formulation code. Specifically, we randomly select three (textual description, code) pairs from the *Evolve/Train* dataset as examples, and use a target textual description from *Evolve/Test* as the test input. For each test case, we repeat the experiment 10 times and report the best result. The prompt used in this process is as follows:

---

**Prompt for GPT-4o**

Please generate Python code for the Mixed-Integer Linear Programming problem corresponding to the description below.
{target desc}

Sample description 1:
{sample_desc1}
Sample code 1:
{sample_code1}

Sample description 2:
{sample_desc2}
Sample code 2:
{sample_code2}

Sample description 3:
{sample_desc3}
Sample code 3:
{sample_code3}

---

#### C.1.2   'FINETUNED LLAMA 3-8B' BASELINE

We implemented another baseline, Finetuned LLaMA 3-8b, which also takes the textual description of a MILP problem as input and generates the corresponding formulation code. This baseline is evaluated on the *Evolve/Test* dataset to assess the performance of the fine-tuned model. We use LLaMA 3-8b-instruct (Dubey et al., 2024) as the base model and perform supervised fine-tuning (SFT). The SFT dataset is constructed using all samples from the *Evolve/Train*, where each sample is a (textual description, formulation code) pair in the following format:

---

**SFT Data Format**

```
1  messages = [
2      {"role": "system", "content": "You are an expert in Mixed-
          Integer Linear Programming."},
3      {"role": "user", "content": "Please generate Python code for
          the Mixed-Integer Linear Programming problem
          corresponding to the description below. \n" +
          description},
4      {"role": "assistant", "content": code}
5  ]
```

---

During testing, we use the same user prompt as input and feed the code generated by the fine-tuned model into GPT-4o for validation, ensuring the output code is free of syntax errors. The prompt used for code checking is as follows:

> **Prompt for Code Checking**
>
> Identify and fix the errors in this code, then output the complete corrected code.
>
> {code}

We perform full-parameter fine-tuning on LLaMA-3-8b-instruct using the XTuner framework (Contributors, 2023), with the hyperparameters listed in Table 7. Fine-tuning is conducted on 8 Nvidia H100 GPUs and takes approximately 6 hours. During testing, we also repeat each experiment 10 times and report the best result.

Table 7: Hyperparameter of Finetuning LLaMA-3-8b.

| Name | Value | Name | Value |
|---|---|---|---|
| Epoch Num | 8 | Learning Rate | 2e-5 |
| Batch Size | 1 | Accumulate Counts | 16 |

### C.1.3 'BOWLY' BASELINE

For the 'Bowly' baseline, we use the official implementation from `https://github.com/simonbowly/mip-generators`, which generates MILP instances based on several specified statistical indicators (e.g., coefficient matrix density, fraction violation rate, etc.). To provide the required inputs, we wrote a script to compute these statistical indicators from the target instances.

### C.1.4 'ACM-MILP' BASELINE

For the 'ACM-MILP' baseline, we use the official implementation provided at `https://github.com/Thinklab-SJTU/ACM-MILP`. For each type of MILP problem, we generate 20 instances to serve as both the training set and the target instances. The trained model is then used to reconstruct these 20 problems. We adopt the same hyperparameters as those used for the preset 'CA' problem, and set the reconstruction ratio to 0.1. It is important to note that ACM-MILP does not guarantee the feasibility of the generated problems—for example, in our experiments, the instances generated for TSP and VRP were infeasible.

### C.2 DETAILS OF *stat metric*

Table 8: Evaluation metrics used in similarity comparison.

| Name | Explanation |
|---|---|
| coef_dens | Fraction of non-zero entries in coefficient matrix. |
| cons_degree_mean | Mean degree of constraint vertices. |
| cons_degree_std | Std of degrees of constraint vertices. |
| var_degree_mean | Mean degree of variable vertices. |
| var_degree_std | Std of degrees of variance vertices. |
| lhs_mean | Mean of non-zero entries in coefficient matrix. |
| lhs_std | Std of non-zero entries in coefficient matrix. |
| rhs_mean | Mean of RHS values. |
| rhs_std | Std of RHS values. |
| modularity | Modularity of the graph. |
| clustering_coef | Clustering coefficient of the graph. |

In previous work (Geng et al., 2023; Guo et al., 2024), graph statistical metrics were used to evaluate the similarity between generated instances and target instances. The full list of metrics is provided

in Table 8. For each individual metric, they calculate the Jensen-Shannon (JS) divergence. Let $JS_i$ denote the JS divergence for the $i^{\text{th}}$ metric. The similarity score for the $i^{\text{th}}$ metric is defined as:

$$\text{score}_i = (\max(JS) - JS_i)/(\max(JS) - \min(JS)). \tag{7}$$

The overall similarity score is the average of all the scores:

$$\text{score} = \frac{1}{11} \sum_{i=1}^{11} \text{score}_i. \tag{8}$$

In our implementation of the *stat metric*, we make two modifications to the above method. First, we remove the *clustering coefficient* metric, as it is always zero for bipartite graphs. Second, we adapt the metric to compute pairwise similarity rather than comparing entire distributions. Existing approaches rely on JS divergence, which is only suitable for comparing two sufficiently large sets of instances. However, in our experimental setting, each group of instances is relatively small—sometimes as few as five instances (e.g., when generating a specific problem class from MIPLIB). In such cases, computing JS divergence leads to high variance.

To address this, we instead use the Jaccard similarity, defined as followswhere $\text{stat}_i$ represents the $i$-th statistical indicator:

$$\text{StatMetric}(p, q) = \frac{1}{10} \sum_{i=1}^{10} \frac{\min(\text{Stat}_i(p), \text{Stat}_i(q))}{\max(\text{Stat}_i(p), \text{Stat}_i(q))},$$
$$\text{StatMetric}(P, Q) = \frac{1}{|P||Q|} \sum_{p \in P} \sum_{q \in Q} \text{StatMetric}(p, q). \tag{9}$$

### C.3 Details of Dataset

In the experiments corresponding to Tables 1 and 2, we used two types of datasets. The first type includes the first four MILP classes from *Evolve/Test* (FCNF, TSP, GA, VRP). For each problem class, we generated 20 instances, which served both as target instances and as training/testing data for ACM-MILP. For the three datasets from MIPLIB (Nursesched, CVS, IIS), we used all available instances provided by MIPLIB as target instances for MILP-Retrieval, and also as training/testing data for ACM-MILP. The dataset statistics are summarized in Table 9.

Table 9: Dataset Statistics of Targeted MILP Instance Generation Experiment.

| Problem Source | Problem Class | Instance Num. | Average $|\mathcal{V}|$ | Average $|\mathcal{C}|$ | Average $|\mathcal{E}|$ |
|---|---|---|---|---|---|
| Evolve/Test | FCNF | 20 | 1096 | 594 | 2192 |
| | TSP | 20 | 1604 | 1567 | 7592 |
| | GA | 20 | 125000 | 750 | 250000 |
| | VRP | 20 | 1088 | 1153 | 7168 |
| MIPLIB | NurseSched | 5 | 19501 | 7231 | 373018 |
| | CVS | 5 | 2536 | 3397 | 9150 |
| | IIS | 2 | 256 | 7551 | 99552 |

## D Extensive Experiment Results

### D.1 More Results on *embedding metric*

We conducted a large-scale experiment to evaluate the performance of the embedding metric on MIPLIB (Gleixner et al., 2021). In the MIPLIB Collection Set, most instances is labeled with a *Group* tag, where instances sharing the same tag are considered to belong to the same problem class. We filtered the Collection Set to include only those groups where every instance has a *Group* tag and can produce a feasible solution within 100 seconds, in order to exclude ultra-scale instances.

Then we filter the group with only one instance. After this filtering process, we obtained 99 classes comprising a total of 361 instances from the original 1,065 instances in the MIPLIB Collection Set (including previous used Nursesched, CVS, IIS classes). All filtered classes and instances are listed in Table 10.

Table 10: Filtered Classes and Instances from MIPLIB Collection Set.

| MILP Class | Instances |
|---|---|
| prod | prod1, prod2 |
| rococo | rococoB10-011000, rococoC11-010100, rococoC12-010001, rococoC10-001000, rococoC11-011100 |
| iis | iis-hc-cov, iis-glass-cov |
| sing | sing326, sing44, sing11, sing5, sing17 |
| shipschedule | shipschedule8shipsmixuci, shipschedule6shipsmixi, shipschedule3shipsi |
| blp | blp-ar98, blp-ir98, blp-ic97, blp-ic98 |
| vpp | vpphard2, vpphard |
| diameterc | diameterc-mstc-v20a190d5i, diameterc-msts-v40a100d5i |
| tanglegram | tanglegram6, tanglegram4 |
| pr-product | p200x1188c, sp150x300d, r50x360, p500x2988, p500x2988d, p500x2988c |
| mine | mine-166-5, mine-90-10 |
| momentum | momentum1, momentum2, momentum3 |
| opm2 | opm2-z6-s1, opm2-z10-s4, opm2-z8-s0, opm2-z12-s8, opm2-z7-s8 |
| acc-tight | acc-tight2, acc-tight4, acc-tight5 |
| map | map14860-20, map10, map06, map18, map16715-04 |
| sp9 | sp98ir, sp98ic, sp98ar, sp97ar, sp97ic |
| ab | ab71-20-100, ab51-40-100, ab69-40-100, ab72-40-100, ab67-40-100 |
| bppc | bppc8-02, bppc6-06, bppc8-09, bppc6-02, bppc4-08 |
| gmu | gmu-35-40, gmut-76-40, gmut-76-50, gmut-75-50, gmu-35-50 |
| dws | dws012-02, dws008-01, dws008-03, dws012-03, dws012-01 |
| decomp | decomp2, decomp1 |
| eil | eilA101-2, eilC76-2, eil33-2 |
| gasprod | gasprod2-1, gasprod1-3, gasprod1-1, gasprod1-2, gasprod2-2 |
| ran | ran12x21, ran13x13, ran14x18-disj-8 |
| eva1aprime | eva1aprime5x5opt, eva1aprime6x6opt |
| assign | assign1-5-8, assign1-10-4 |
| ofi | ofi, ofi2 |
| fhnw-schedule | fhnw-schedule-paira400, fhnw-schedule-paira200, fhnw-schedule-pairb200, fhnw-schedule-paira100, fhnw-schedule-pairb400 |
| cmflsp | cmflsp50-24-8-8, cmflsp50-24-10-4, cmflsp40-24-10-7, cmflsp40-36-2-10, cmflsp60-36-2-6 |
| csched | csched008, csched007, csched010 |
| network_design | germany50-UUM, cost266-UUE, ta2-UUE, dfn-bwin-DBE, ta1-UUM |
| ger50 | ger50-17-ptp-pop-6t, ger50-17-trans-dfn-3t, ger50-17-trans-pop-3t, ger50-17-ptp-pop-3t, ger50_17_trans |
| fastxgemm | fastxgemm-n2r6s0t2, fastxgemm-n3r21s3t6, fastxgemm-n3r22s4t6, fastxgemm-n3r23s5t6, fastxgemm-n2r7s4t1 |
| triptim | triptim8, triptim4, triptim7, triptim1, triptim2 |
| gen-ip | gen-ip002, gen-ip021, gen-ip054, gen-ip036, gen-ip016 |
| snp | snp-10-004-052, snp-02-004-104, snp-10-052-052, snp-04-052-052, snp-06-004-052 |
| satellites | satellites2-40, satellites3-25, satellites2-60-fs, satellites2-25, satellites4-25 |
| rmatr | rmatr200-p10, rmatr100-p5, rmatr200-p5, rmatr100-p10, rmatr200-p20 |
| dano | dano3mip, dano3_5, danoint, dano3_3 |
| cvrp | cvrpp-n16k8vrpi, cvrpa-n64k9vrpi, cvrpb-n45k5vrpi, cvrpsimple2i |

Continued from previous page

| MILP Class | Instances |
|---|---|
| f2gap | f2gap401600, f2gap201600, f2gap801600, f2gap40400 |
| sorrell | sorrell4, sorrell3, sorrell8, sorrell7 |
| uccase | uccase8, uccase12, uccase9, uccase10, uccase7 |
| nu120 | nu120-pr9, nu120-pr12 |
| nursesched | nursesched-medium04, nursesched-medium-hint03, nursesched-sprint-hidden09, nursesched-sprint-late03, nursesched-sprint02 |
| berlin | berlin_5_8_0, berlin |
| nxy-z | n6-3, n13-3, n7-3, n5-3, n9-3 |
| tbfp | tbfp-bigm, tbfp-network |
| nh97 | nh97_tension, nh97_potential |
| qnet | qnet1, qnet1_o |
| markshare | markshare1, markshare_4_0, markshare_5_0, markshare2 |
| timtab | timtab1, timtab1CUTS |
| swath | swath3, swath2, swath1, swath |
| app | app2-1, app1-1, app3, app2-2, app1-2 |
| core | core2586-950, core4872-1529, core4284-1064, core2536-691 |
| allcolor | allcolor58, allcolor10 |
| reblock | reblock166, reblock420, reblock354, reblock115 |
| pizza | pizza78i, pizza27i |
| roi | roi5alpha10n8, roi2alpha3n4 |
| graphdraw | graphdraw-gemcutter, graphdraw-grafo2, graphdraw-opmanager, graphdraw-mainerd, graphdraw-domain |
| genus | genus-sym-g31-8, genus-sym-grafo5708-48, genus-sym-g62-2, genus-g31-8, genus-g61-25 |
| ex | ex9, ex1010-pi, ex10, exp-1-500-5-5 |
| nexp | nexp-150-20-1-5, nexp-150-20-8-5, nexp-50-20-4-2, nexp-50-20-1-1 |
| aflow | aflow40b, aflow30a |
| splice | splice1k1i, splice1k1 |
| fcnf | g200x740, h80x6320, h80x6320d, k16x240b, h50x2450 |
| pigeon | pigeon-10, pigeon-08, pigeon-13, pigeon-20, pigeon-16 |
| lectsched | lectsched-1, lectsched-4-obj, lectsched-5-obj, lectsched-3, lectsched-2 |
| adult | adult-regularized, adult-max5features |
| xmas | xmas10, xmas10-2 |
| shiftreg | shiftreg1-4, shiftreg2-7, shiftreg5-1 |
| beasley | beasleyC2, beasleyC3, beasleyC1 |
| seymour | seymour1, seymour |
| cvs | cvs16r89-60, cvs16r128-89, cvs08r139-94, cvs16r106-72, cvs16r70-62 |
| nseq | n2seq36f, n3seq24, n2seq36q, n3div36 |
| k1mushroom | k1mushroomi, k1mushroom |
| mc | mc7, mc8, mc11 |
| traininstance | traininstance2, traininstance6 |
| sct | sct2, sct31, sct32, sct5, sct1 |
| tpl-tub | tpl-tub-ws1617, tpl-tub-ss16 |
| mas | mas76, mas74 |
| gsvm | gsvm2rl9, gsvm2rl3, gsvm2rl5, gsvm2rl12, gsvm2rl11 |
| physiciansched | physiciansched5-3, physiciansched6-1, physiciansched3-4, physiciansched3-3, physiciansched6-2 |
| bienst | bienst1, bienst2 |
| drayage | drayage-100-12, drayage-25-23, drayage-25-27, drayage-25-32, drayage-100-23 |
| milo | milo-v12-6-r1-58-1, milo-v12-6-r1-75-1, milo-v13-4-3d-3-0, milo-v12-6-r2-40-1, milo-v13-4-3d-4-0 |
| leo | leo2, leo1 |

Continued from previous page

| MILP Class | Instances |
|---|---|
| set3 | set3-16, set3-09, set3-10, set3-15, set3-20 |
| radiation | radiationm18-12-05, radiationm40-10-02 |
| chromaticindex | chromaticindex128-5, chromaticindex256-8, chromaticindex512-7, chromaticindex32-8, chromaticindex1024-7 |
| air | air03, air05, air04 |
| graph | graph20-80-1rand, graph40-20-1rand, graph20-20-1rand, graph40-40-1rand, graph40-80-1rand |
| n37 | n370b, n3700, n3707, n3709, n3705 |
| 30_70_45 | 30_70_45_095_98, 30_70_45_05_100, 30_70_45_095_100 |
| bley | bley_xs2, bley_xl1, bley_xs1, bley_xs1noM |
| bmocbd | bmocbd2, bmocbd3, bmocbd |
| piperout | piperout-03, piperout-08, piperout-27, piperout-d20, piperout-d27 |
| hgms | hgms62, hgms-det, hgms30 |
| mspsp | mspsphard03i, mspsphard01i |

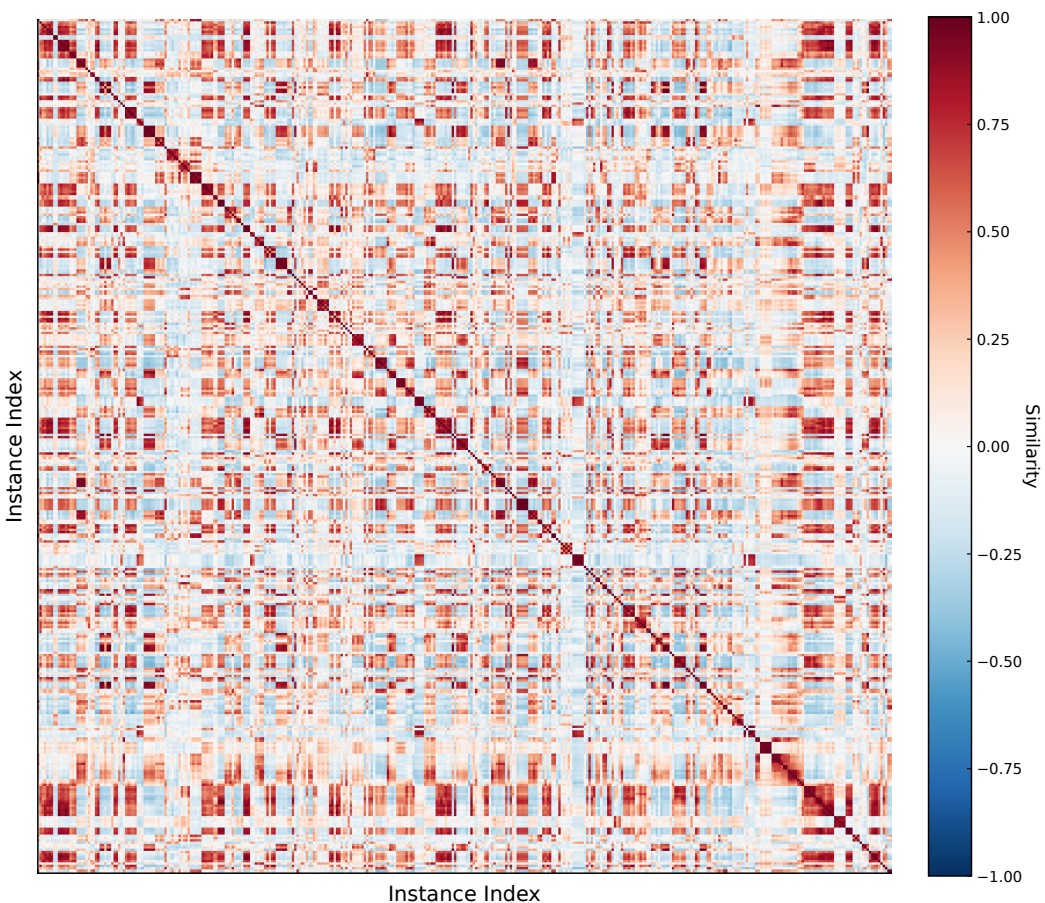

Figure 15: Similarity Matrix of *embedding metric* on 361 instances (99 classes) from MIPLIB.

We computed the pairwise similarity among the 361 instances using the *embedding metric*. The instance indices follow the same order as presented in Table 10. The results are visualized in Figure 15. In the figure, red indicates higher similarity while blue indicates lower similarity. Since the indices of instances from the same problem class are placed consecutively, we observe that many red-colored squares appear along the diagonal of the similarity matrix. This suggests that different

instances from the same class—often with significantly varying sizes—can still yield high similarity scores under the *embedding metric*. Moreover, there are also numerous red squares off the diagonal (though generally with lower similarity than those on the diagonal), indicating that the *embedding metric* is capable of discovering related classes within MIPLIB.

The experimental results demonstrate that the *embedding metric* effectively distinguishes unseen instances from MIPLIB, providing strong evidence of its robustness.

## D.2 MORE RESULTS ON MIPLIB

We further evaluate the performance of MILP-Retrieval on the exact same dataset as listed in Table 10. We first visualize the distribution of number of variables/constraints of the instances in Figure 16, showing that compared to the Evolve/Train visualized in Figure 11, the filtered MIPLIB has a greater diversity.

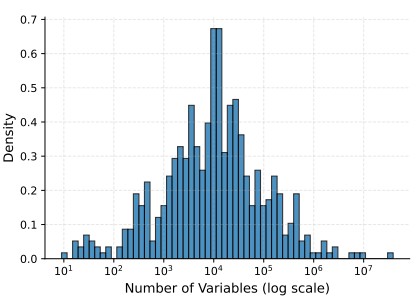
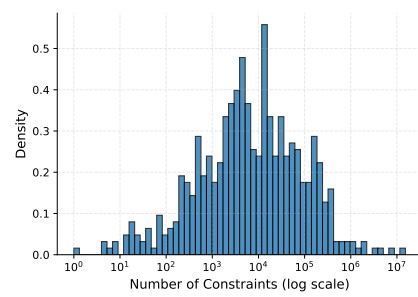

(a) Distribution of Number of Variables.  (b) Distribution of Number of Constraints.

Figure 16: Visualization of All Instances in filtered MIPLIB.

We then evaluate MILP-Retrieval on the same dataset, results are presented in Table 11. For each problem class, we report the similarity between the target instances and the generated instances, using both *embedding metric* and *stat metric*. The results are presented in Table 11. It is worth highlighting that across all MILP classes, the average *embedding metric* is 0.701, while the average *stat metric* is 0.236. These results indicate that in most cases, MILP-Retrieval is able to generate instances with relatively high *embedding metric* to the corresponding target instances.

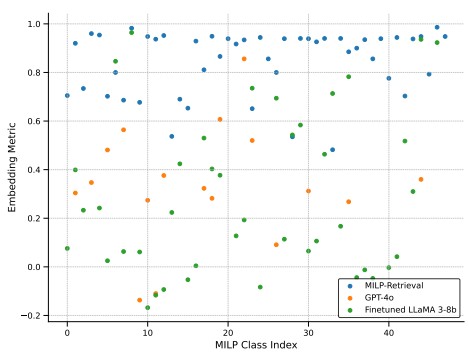
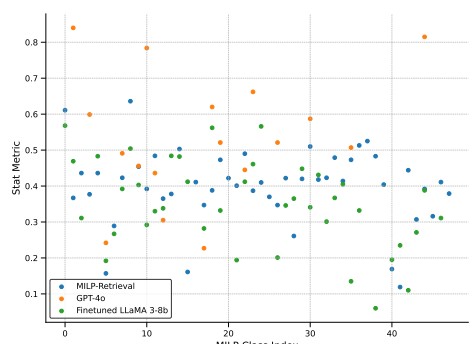

Figure 17: Comparison on *embedding metric*.

Figure 18: Comparison on *stat metric*.

Table 12: Feasible Rate of Generated Formulation Code.

| Method | 1-shot | 4-shots | 10-shots |
|---|---|---|---|
| GPT-4o | 2/50 | 6/50 | 17/50 |
| Finetuned LLaMA 3-8b | 14/50 | 26/50 | 42/50 |

Table 11: Feasible Rate of Generated Formulation Code.

| MILP Class | embedding metric | stat metric | MILP Class | embedding metric | stat metric |
|---|---|---|---|---|---|
| prod | 0.81 ± 0.039 | 0.122 ± 0.02 | rococo | 0.854 ± 0.055 | 0.17 ± 0.089 |
| iis | 0.829 ± 0.046 | 0.234 ± 0.003 | sing | 0.911 ± 0.042 | 0.23 ± 0.05 |
| shipschedule | 0.876 ± 0.035 | 0.113 ± 0.027 | blp | 0.862 ± 0.078 | 0.096 ± 0.005 |
| vpp | 0.418 ± 0.513 | 0.312 ± 0.019 | diameterc | 0.81 ± 0.092 | 0.1 ± 0.015 |
| tanglegram | 0.817 ± 0.071 | 0.115 ± 0.027 | pr-product | 0.711 ± 0.404 | 0.271 ± 0.072 |
| mine | 0.679 ± 0.157 | 0.269 ± 0.034 | momentum | 0.426 ± 0.312 | 0.246 ± 0.054 |
| opm2 | 0.861 ± 0.053 | 0.296 ± 0.043 | acc-tight | 0.914 ± 0.036 | 0.497 ± 0.014 |
| map | 0.93 ± 0.03 | 0.111 ± 0.003 | sp9 | 0.481 ± 0.341 | 0.324 ± 0.035 |
| ab | 0.668 ± 0.094 | 0.099 ± 0.001 | bppc | 0.667 ± 0.169 | 0.225 ± 0.024 |
| gmu | 0.92 ± 0.025 | 0.289 ± 0.034 | dws | 0.762 ± 0.054 | 0.106 ± 0.007 |
| decomp | 0.593 ± 0.246 | 0.46 ± 0.103 | eil | 0.97 ± 0.019 | 0.212 ± 0.022 |
| gasprod | 0.523 ± 0.26 | 0.171 ± 0.04 | ran | 0.715 ± 0.236 | 0.236 ± 0.069 |
| eva1aprime | 0.772 ± 0.17 | 0.324 ± 0.023 | assign | 0.67 ± 0.128 | 0.12 ± 0.009 |
| ofi | 0.814 ± 0.045 | 0.142 ± 0.002 | fhnw-schedule | 0.751 ± 0.081 | 0.277 ± 0.052 |
| cmflsp | 0.931 ± 0.021 | 0.311 ± 0.031 | csched | 0.51 ± 0.563 | 0.247 ± 0.026 |
| network_design | 0.304 ± 0.358 | 0.274 ± 0.079 | ger50 | 0.376 ± 0.346 | 0.235 ± 0.038 |
| fastxgemm | 0.896 ± 0.066 | 0.064 ± 0.021 | triptim | 0.714 ± 0.326 | 0.183 ± 0.067 |
| gen-ip | 0.249 ± 0.342 | 0.167 ± 0.065 | snp | 0.799 ± 0.096 | 0.322 ± 0.008 |
| satellites | 0.565 ± 0.259 | 0.12 ± 0.079 | rmatr | 0.809 ± 0.101 | 0.252 ± 0.038 |
| dano | 0.771 ± 0.099 | 0.324 ± 0.056 | cvrp | 0.86 ± 0.093 | 0.429 ± 0.07 |
| f2gap | 0.467 ± 0.11 | 0.118 ± 0.056 | sorrell | 0.909 ± 0.037 | 0.301 ± 0.07 |
| uccase | 0.092 ± 0.383 | 0.391 ± 0.056 | nu120 | 0.875 ± 0.033 | 0.153 ± 0.018 |
| nursesched | 0.883 ± 0.085 | 0.231 ± 0.076 | berlin | 0.388 ± 0.61 | 0.48 ± 0.146 |
| nxy-z | 0.746 ± 0.046 | 0.226 ± 0.04 | tbfp | 0.414 ± 0.541 | 0.247 ± 0.207 |
| nh97 | 0.212 ± 0.474 | 0.08 ± 0.039 | qnet | 0.845 ± 0.083 | 0.285 ± 0.002 |
| markshare | 0.669 ± 0.288 | 0.218 ± 0.094 | timtab | 0.66 ± 0.215 | 0.312 ± 0.047 |
| swath | 0.881 ± 0.027 | 0.109 ± 0.002 | app | 0.307 ± 0.489 | 0.23 ± 0.064 |
| core | 0.818 ± 0.076 | 0.122 ± 0.005 | allcolor | 0.82 ± 0.075 | 0.168 ± 0.019 |
| reblock | 0.81 ± 0.077 | 0.326 ± 0.036 | pizza | 0.914 ± 0.037 | 0.298 ± 0.014 |
| roi | 0.438 ± 0.329 | 0.115 ± 0.04 | graphdraw | 0.544 ± 0.246 | 0.235 ± 0.046 |
| genus | 0.943 ± 0.027 | 0.34 ± 0.029 | ex | 0.098 ± 0.513 | 0.305 ± 0.158 |
| nexp | 0.344 ± 0.462 | 0.382 ± 0.063 | aflow | 0.874 ± 0.044 | 0.322 ± 0.022 |
| splice | 0.89 ± 0.077 | 0.195 ± 0.051 | fcnf | 0.361 ± 0.482 | 0.291 ± 0.123 |
| pigeon | 0.764 ± 0.143 | 0.202 ± 0.036 | lectsched | 0.601 ± 0.104 | 0.306 ± 0.007 |
| adult | 0.923 ± 0.013 | 0.177 ± 0.024 | xmas | 0.734 ± 0.245 | 0.159 ± 0.005 |
| shiftreg | 0.728 ± 0.083 | 0.16 ± 0.006 | beasley | 0.985 ± 0.005 | 0.436 ± 0.041 |
| seymour | 0.385 ± 0.475 | 0.106 ± 0.001 | cvs | 0.814 ± 0.078 | 0.430 ± 0.121 |
| nseq | 0.361 ± 0.315 | 0.142 ± 0.023 | k1mushroom | 0.767 ± 0.195 | 0.246 ± 0.086 |
| mc | 0.97 ± 0.015 | 0.362 ± 0.009 | traininstance | 0.86 ± 0.021 | 0.269 ± 0.023 |
| sct | 0.491 ± 0.192 | 0.216 ± 0.034 | tpl-tub | 0.778 ± 0.053 | 0.267 ± 0.002 |
| mas | 0.649 ± 0.053 | 0.174 ± 0.015 | gsvm | 0.65 ± 0.172 | 0.098 ± 0.032 |
| physiciansched | 0.723 ± 0.408 | 0.271 ± 0.046 | bienst | 0.945 ± 0.018 | 0.196 ± 7.377 |
| drayage | 0.957 ± 0.014 | 0.293 ± 0.035 | milo | 0.917 ± 0.043 | 0.271 ± 0.014 |
| leo | 0.762 ± 0.022 | 0.131 ± 0.006 | set3 | 0.802 ± 0.05 | 0.432 ± 0.014 |
| radiation | 0.579 ± 0.069 | 0.282 ± 0.03 | chromaticindex | 0.662 ± 0.09 | 0.148 ± 0.025 |
| air | 0.866 ± 0.066 | 0.163 ± 0.067 | graph | 0.919 ± 0.065 | 0.239 ± 0.157 |
| n37 | 0.761 ± 0.036 | 0.477 ± 0.002 | 30_70_45 | 0.94 ± 0.044 | 0.295 ± 0.106 |
| bley | 0.481 ± 0.398 | 0.134 ± 0.074 | bmocbd | 0.882 ± 0.108 | 0.405 ± 0.012 |
| piperout | 0.68 ± 0.129 | 0.221 ± 0.104 | hgms | 0.057 ± 0.465 | 0.066 ± 0.002 |
| mspsp | 0.908 ± 0.015 | 0.373 ± 0.012 | | | |

### D.3 More Results on *Evolve/Test* Dataset

We provide the experimental results of MILP-Retrieval and two LLM-based baselines(GPT-4o, Finetuned LLaMA 3-8b) on the full Evolve/Test dataset, which includes 50 MILP classes. The LLM-based baselines are evaluated by directly inputting the textual description of the target instance. The LLM-based baselines are tested under a 10-shot setting, where each experiment is repeated 10 times, and the best result is reported. In the figure, missing entries for the LLM-based baselines indicate that the method failed to generate a feasible formulation code for those instances.

This experiment serves as a supplement to the Evolve/Test results presented in Tables 1 and 2, with the results shown in Figure 17 and 18. Additionally, we report the proportion of successful formulation code generations by the LLM-based baselines across different numbers of trials, as presented in Table 12. These results demonstrate that MILP-Retrieval maintains strong performance even on larger-scale datasets, highlighting its robustness.

### D.4 Details and Additional Results on Enhancing Learning-based Solver

#### D.4.1 Introduction of Underlying Learning-based Solver

**Neural Diving** (Nair et al., 2020) is a machine learning approach for solving MILP problems that focuses on generating high-quality joint variable assignments. It trains a GNN to produce multiple partial assignments for the variables within a MILP instance. These partial assignments effectively define smaller, more manageable sub-MILPs. These sub-MILPs, with their reduced complexity due to many variables being fixed, are then solved using a standard MILP solver, such as SCIP, to complete the assignments and construct high-quality solutions.

**Learn-to-Branch** (Gasse et al., 2019) is an imitation-learning approach to improve variable selection in branch-and-bound for MILPs. It represents each MILP as a bipartite graph and encodes solver states with rich node and edge features. A lightweight graph convolutional neural network performs message passing between variables and constraints, producing scores used to choose branching variables. The model is trained via behavioral cloning to imitate strong branching decisions using a cross-entropy loss. This GCNN architecture reduces feature engineering, is efficient at inference time, and generalizes to larger problem instances while outperforming existing learning-based and hand-crafted branching rules.

**Predict-and-Search** (Han et al., 2023) proposes a GNN-guided framework for solving MILPs more efficiently. First, each MILP instance is encoded as a bipartite graph, and a graph neural network is trained via supervised distribution learning to predict marginal probabilities for all binary variables, indicating how likely each variable is to take value 1 in high-quality solutions. Instead of fixing variables directly, the method constructs a trust region neighborhood around a partial solution derived from these predictions. A MILP solver then searches within this restricted region to find a high-quality feasible solution. This approach improves solution quality over SCIP, Gurobi, and fixing-based neural methods.

We used a third-party implementation of these frameworks provided by `https://github.com/thuiar/MILPBench`, which network structure is exactly the same as described.

#### D.4.2 Additional Results on Enhancing Learning-based Solver

To simulate a data-scarce setting, we randomly generate 5 instances per problem class to serve as the training set, and 15 instances as the test set. These 5 instances also act as the target instances for MILP-Retrieval, as well as the training data for other learning-based baseline methods.

For Neural Diving, we firstly conduct an ablation study on the number of instances generated by each method, with the results shown in Table 13, 14, 15 and 16. We mark '-' for the cases where the original method cannot generate a feasible instance, and we mark 'infeasible' for the cases where Neural Diving cannot find a feasible solution (that is, the predicted partial solution has violated the constraints). These results demonstrate that MILP-Retrieval can effectively enhance the performance of Neural Diving and, and in most cases it outperforms the baseline methods.

Table 13: Neural Diving Experimental results on **FCNF** problem with respect to the number of problems generated by each method. We use each method to generate different numbers (10, 20, 40, 80) of instances and add them to the training set.

| FCNF | Raw | MILP-Retrieval | ACM-MILP | GPT-4o | Finetuned LLaMA 3-8b |
|------|-----|----------------|----------|--------|----------------------|
| 10 | | 1456.77 ± 244.39 | 1888.65 ± 266.46 | - | 1299.69 ± 205.35 |
| 20 | 1604.77 ± 311.27 | 1266.71 ± 190.74 | 1767.03 ± 245.33 | - | 1160.32 ± 168.10 |
| 40 | | 1117.14 ± 187.68 | 1520.08 ± 200.00 | - | 1228.64 ± 399.68 |
| 80 | | 1612.60 ± 249.79 | 1781.44 ± 253.04 | - | 1096.72 ± 182.26 |

Table 14: Neural Diving Experimental results on **TSP** problem with respect to the number of problems generated by each method. We use each method to generate different numbers (10, 20, 40, 80) of instances and add them to the training set.

| FCNF | Raw | MILP-Retrieval | ACM-MILP | GPT-4o | Finetuned LLaMA 3-8b |
|------|-----|----------------|----------|--------|----------------------|
| 10 | | 891.53 ± 83.79 | - | 890.79 ± 82.84 | 895.40 ± 82.55 |
| 20 | 944.85 ± 98.45 | 891.53 ± 83.79 | - | infeasible | infeasible |
| 40 | | 891.47 ± 83.86 | - | 891.53 ± 83.79 | 924.40 ± 86.83 |
| 80 | | 935.13 ± 94.76 | - | infeasible | 991.2 ± 92.46 |

Table 15: Neural Diving Experimental results on **GA** problem with respect to the number of problems generated by each method. We use each method to generate different numbers (10, 20, 40, 80) of instances and add them to the training set.

| GA | Raw | MILP-Retrieval | ACM-MILP | GPT-4o | Finetuned LLaMA 3-8b |
|----|-----|----------------|----------|--------|----------------------|
| 10 | | -49989.60 ± 4.98 | infeasible | - | -49999.93 ± 0.26 |
| 20 | -49768.53 ± 53.77 | -49999.93 ± 0.26 | infeasible | - | infeasible |
| 40 | | -49991.25 ± 4.92 | infeasible | - | -49293.00 ± 15.23 |
| 80 | | -49997.53 ± 1.41 | infeasible | - | -49129.60 ± 13.40 |

Table 16: Neural Diving Experimental results on **VRP** problem with respect to the number of problems generated by each method. We use each method to generate different numbers (10, 20, 40, 80) of instances and add them to the training set.

| VRP | Raw | MILP-Retrieval | ACM-MILP | GPT-4o | Finetuned LLaMA 3-8b |
|-----|-----|----------------|----------|--------|----------------------|
| 10 | | infeasible | - | infeasible | - |
| 20 | 911.20 ± 91.02 | infeasible | - | infeasible | - |
| 40 | | 774.91 ± 52.64 | - | 827.65 ± 40.98 | - |
| 80 | | 773.08 ± 44.13 | - | infeasible | - |

For Predict-and-Search, when collecting training data, we set a maximum solving time of 3600 seconds for each problem and gather 500 solution trajectories for training. The data is split into training and validation sets with a 4:1 ratio, and Gurobi 12.0 is used as the solver. In Table 17, we report the average solving time on the test set as well as the optimality gap (in percentage) between the obtained solutions and the ground-truth optimal solutions.

Table 17: The result of Predict-and-Search framework. We reported the average solution time on the test set, and the values in parentheses are the gaps between the obtained solutions and the optimal solution to the problem.

| | Raw | MILP-Retrieval | ACM-MILP | GPT-4o | Finetuned LLaMA 3-8b |
|------|-----|----------------|----------|--------|----------------------|
| FCNF | 150.40 (0.0689) | **147.32** (0.0689) | 150.45 (0.0689) | - | 148.21 (0.0689) |
| TSP | 0.783 (0) | **0.767** (0) | - | 0.755 (0) | 0.769 (0) |
| GA | 36.46 (0) | **35.52** (0) | 36.30 (0) | - | 36.51 (0) |
| VRP | 222.67 (0.00974) | **215.35** (0.00974) | - | 221.60 (0.00974) | - |

For Learn-to-Branch, we use the same training/testing data split and also limit data collection for each problem to 3600 seconds. Evaluation is conducted with SCIP 8.0.3, which is compatible with Ecole 0.8.1. Table 18 reports the results for Learn-to-Branch.

Table 18: The result of Learn-to-Branch framework. We reported the average solution time on the test set.

|      | Raw    | MILP-Retrieval | ACM-MILP | GPT-4o   | Finetuned LLaMA 3-8b |
|------|--------|----------------|----------|----------|----------------------|
| FCNF | 240.33 | **235.38**     | 236.24   | -        | 244.13               |
| TSP  | 15.87  | **15.22**      | -        | 15.05    | 15.54                |
| GA   | 36.65  | **36.38**      | 36.55    | -        | 36.55                |
| VRP  | 353.49 | 355.57         | -        | **351.20** | -                  |

Overall, across both tasks, we observe that in most cases the problems generated by MILP-Retrieval strengthen solver performance under data-scarce settings and outperform existing baselines.

## E  DISCUSSIONS

**In this section, we present a more in-depth discussion of our work in a Q&A format.**

**Q1**: How can we ensure that MILP-Retrieval consistently retrieves sufficiently similar instances for any given MILP problem?

Our method does not guarantee this for all MILP instances. However, it is important to note that the three components of our approach—the retrieve-and-generate paradigm, the formulation code library, and the MILP embedding model—are decoupled. This design allows future advancements in either the construction of more diverse MILP libraries or the development of improved MILP embedding models to be directly integrated into our framework. In this work, we employ the existing MILP-Evolve method to generate a diverse MILP library and provide extensive empirical evidence demonstrating the viability of our retrieve-and-generate paradigm.

**Q2**: What are the broader connections between this work and the field of machine learning?

*Reverse image search* has been a significant research topic in computer vision and machine learning in recent years, with applications such as finding similar images, identifying image sources, and retrieving relevant information about images. Analogously, our proposed framework, MILP-Retrieval, can be seen as enabling *reverse MILP search*—a powerful retrieval tool tailored to the domain of MILP problems, which encompass the majority of combinatorial optimization problems. We also demonstrate the effectiveness of this tool in downstream tasks that aim to enhance learning-based MILP solvers.

**Q3**: Is there a risk of bias in the MILP embedding model?

Yes, since all embedding models are trained on finite datasets, they inevitably carry some bias and cannot perfectly capture the distribution of real-world data. To mitigate this issue, we utilize the state-of-the-art method for generating MILP libraries to construct a sufficiently large and diverse dataset, helping to reduce the impact of bias on the embedding model.

**Q4**: Why does MILP-Retrieval perform poorly under the *stat metric*, and why does ACM-MILP perform poorly under the *embedding metric*?

The MILP embedding model implemented in our work is aligned with the semantic structure of the problem, rather than its size. As shown in Figure 4, the statistical metric is sensitive to size differences among instances within the same problem category, whereas the embedding metric emphasizes structural and semantic similarities. MILP-Retrieval aims to retrieve instances that are semantically similar to the target instance, which does not necessarily ensure similarity in size—resulting in lower scores on the stat metric. In contrast, ACM-MILP reconstructs parts of the original problem while preserving its size, but this can alter the semantic content, leading to poorer performance on the embedding metric.

**Q5**: Why does MILP-Retrieval underperform compared to LLM-based baselines in some experiments of enhancing learning-based solver?

In our experiments, the LLM-based baselines (GPT-4o, Finetuned LLaMA 3-8b) generate formulation code based on textual descriptions of the problem. They are not capable of directly generating formulation code from target instances, and therefore can only be evaluated on synthetic datasets like Evolve/Test, which include textual descriptions, but not on real-world benchmarks such as MIPLIB. As such, these experimental results are not fully comparable. We opted to implement the LLM-based baselines using textual descriptions because, to the best of our knowledge, there are currently no available Graph-Language Models (GLMs) capable of jointly processing graph-structured data and natural language inputs.

**Q6**: Why does MILP-Retrieval underperform compared to LLM-based baselines in some experiments of enhancing learning-based solver?

In our experiments, the LLM-based baselines (GPT-4o, Finetuned LLaMA 3-8b) generate formulation code based on textual descriptions of the problem. They are not capable of directly generating formulation code from target instances, and therefore can only be evaluated on synthetic datasets like Evolve/Test, which include textual descriptions, but not on real-world benchmarks such as MIPLIB. As such, these experimental results are not fully comparable. We opted to implement the LLM-based baselines using textual descriptions because, to the best of our knowledge, there are currently no available Graph-Language Models (GLMs) capable of jointly processing graph-structured data and natural language inputs.

