# OpenReview forum: "Targeted MILP Instance Generation via Formulation Code Retrieval"
_ICLR.cc/2026/Conference — Submitted to ICLR 2026_

### Official Review · Reviewer_5Bjq · 2025-10-26

**Soundness:** 3
**Presentation:** 3
**Contribution:** 3
**Rating:** 6
**Confidence:** 4

**Summary:**

This paper introduces "MILP-Retrieval," a novel framework to address the critical problem of data scarcity for training data-driven Mixed-Integer Linear Programming (MILP) solvers. The authors argue that existing generative methods (e.g., VAEs, diffusion models) are highly inefficient, as they require training a separate, complex model for each distinct problem class and offer poor control over the generated instance's properties.

MILP-Retrieval proposes a paradigm shift, changing the problem from "generation" to "retrieval and tuning." The framework consists of several key components:

MILP Library, MILP Embedding Model, Embedding Metric, Retrieval and Tuning.

**Strengths:**

Novel and Highly Practical Paradigm: The core idea is the paper's greatest strength. It astutely reframes a very difficult "generation" problem into a much more tractable "retrieval-then-tuning" problem. This approach is computationally far more efficient, as the expensive pre-training of the embedding model is a one-time, amortized cost. It avoids the need to train a new generative model for every new problem class.

Excellent Controllability: A significant advantage over all other generative methods. By retrieving the underlying code, the method gains direct, interpretable control over the generation process. The "Targeted Tuning" using Bayesian optimization (Figures 7 and 8) is particularly powerful, demonstrating the ability to generate instances that match a specific difficulty (e.g., target solve time), which is extremely valuable for solver testing and training.

Strong Contribution in MILP Similarity: The paper makes a valuable standalone contribution by proposing the "embedding metric." The comparison in Figure 4 is very clear and convincing. It shows the embedding metric captures the semantic class of an instance (Fig 4c) even when scale varies, whereas the statistical metric (Fig 4d) is completely confounded by scale.

**Weaknesses:**

High Dependency on Library Quality: The entire framework's performance is fundamentally capped by the quality and comprehensiveness of the MILP library. If a target instance belongs to a novel problem class that is not well-represented in the library, the retrieval will fail to find a good match, and the "tuning" step will be useless. The paper acknowledges this, but the risk of "out-of-distribution" failure is significant.

Limitations of "Tuning": The "tuning" mechanism only adjusts parameters (e.g., $N, M$, cost ranges) within a fixed formulation code structure. This is a limitation. If a target instance has a slightly different structural property (e.g., an extra set of constraints) not present in the retrieved code, parameter tuning alone can never reproduce it. This limits the "fineness" of the generation.

**Questions:**

Out-of-Distribution Behavior: What is the method's failure mode? If given a target instance from a problem class that is truly novel and not in the library (e.g., from a completely different domain), what formulation code does it retrieve? Does it retrieve a "least-bad" match that produces nonsensical instances?

Library Sensitivity: Figure 10 shows robustness to library size, but what is the minimum viable library required for this approach to be practical? How much human effort is needed to curate a "good enough" library to cover a broad range of real-world problems?

---

> ### Author Response · Authors · 2025-11-20
> **Response to Reviewer**
>
> We appreciate the valuable comments from the reviewer, and we are willing to address the concerns.
>
> > **W1**. High Dependency on Library Quality: The entire framework's performance is fundamentally capped by the quality and comprehensiveness of the MILP library. If a target instance belongs to a novel problem class that is not well-represented in the library, the retrieval will fail to find a good match, and the "tuning" step will be useless. The paper acknowledges this, but the risk of "out-of-distribution" failure is significant.
>
> We agree that the quality of the formulation-code library affects the quality of generated MILPs. However, the retrieve-then-tune paradigm, which is the core contribution of our paper, is *decoupled* from both the formulation-code library and the embedding model. A future library with greater diversity or higher quality can be seamlessly plugged into our framework.
>
> As described in Appendix B.2, our current formulation-code library is constructed via a cross-evolving and recombination procedure. In principle, this procedure can synthesize an arbitrarily large library. In this work, we generated a library with 4,000 formulation codes, and the generation pipeline is fully ready to scale further.
>
>
>
> > **W2**. Limitations of "Tuning": The "tuning" mechanism only adjusts parameters (e.g. $N$, $M$, cost ranges) within a fixed formulation code structure. This is a limitation. If a target instance has a slightly different structural property (e.g., an extra set of constraints) not present in the retrieved code, parameter tuning alone can never reproduce it. This limits the "fineness" of the generation.
>
> The objective of the tuning step in MILP-Retrieval is not to correct structural dissimilarity; rather, it is designed to control the scale and difficulty of the generated instances. Because of this, our retrieval stage uses an embedding model that is intentionally scale-unaware, ensuring that structurally similar instances are already retrieved in the first place. Under this design, the retrieve step handles structural alignment, and the tuning step adjusts quantitative problem attributes.
>
>
>
> > **Q1**. Out-of-Distribution Behavior: What is the method's failure mode? If given a target instance from a problem class that is truly novel and not in the library (e.g., from a completely different domain), what formulation code does it retrieve? Does it retrieve a "least-bad" match that produces nonsensical instances?
>
> Every formulation code in our library is validated before inclusion. Therefore, even when presented with a *truly novel*target instance, the retrieved formulation code, while potentially having lower similarity, still corresponds to a meaningful and valid MILP formulation.
>
> Moreover, our retrieve-then-tune pipeline provides a natural mechanism for estimating confidence: The retrieval similarity score itself serves as a direct indicator of expected generation quality. This is an advantage over existing MILP generative approaches, which typically rely on *post-hoc evaluation* to assess generated-instance quality.
>
>
>
> > **Q2**. Library Sensitivity: Figure 10 shows robustness to library size, but what is the minimum viable library required for this approach to be practical? How much human effort is needed to curate a "good enough" library to cover a broad range of real-world problems?
>
> We believe that the library used in our experiments, a collection of 4,000 automatically generated formulation codes constitutes a reasonable *minimum viable library*. Our justification is based on real-world evaluations using MIPLIB. As reported in Appendix D.2, across 99 classes and 361 instances from MIPLIB, the instances generated by MILP-Retrieval achieve an average embedding metric of 0.701, demonstrating strong coverage of diverse real-world MILP structures.
>
> Importantly, this formulation code library is not manually curated. It is produced automatically using the synthesis procedure described in Appendix B.2. Generating the entire library required roughly one month and approximately $50 of API cost. We have released the full library in our anonymized repository.

---

### Official Review · Reviewer_x6NP · 2025-10-29

**Soundness:** 3
**Presentation:** 3
**Contribution:** 3
**Rating:** 4
**Confidence:** 4

**Summary:**

This paper proposes MILP-Retrieval, a retrieval-and-tune framework for targeted MILP instance generation. Instead of reconstructing instance structures with a class-specific generative model, the method builds a multi-modal MILP library (instances, formulation code, bipartite graphs, textual descriptions), trains a graph–text contrastive embedding model, uses an embedding-based similarity metric to retrieve the closest formulation code, and then tunes code parameters (randomized or Bayesian/SMAC) to control scale and difficulty before executing the code to synthesize instances. Experiments show higher semantic similarity under the proposed embedding metric, controllable hardness, and downstream gains for Neural Diving across four classes.

**Strengths:**

1. This paper investigates a new generation paradigm. Instead of directly generating problem instances, it retrieves formulation code and then produces new instances by executing and tuning that code. This approach enhances controllability and interpretability while avoiding per-class training required by generative models.

1. The ability to generate meaningful instances on MIPLIB is impressive. Prior methods like VAE-based generators typically focus on synthetic or homogeneous datasets. Demonstrating that retrieval-based synthesis can operate on real-world MIPLIB formulations marks a step forward in practicality.

1. The downstream task improvements are important. The authors test Neural Diving on four datasets (FCNF, TSP, GA, VRP) and show consistent improvement when trained with instances generated by MILP-Retrieval.

**Weaknesses:**

1. The paper is closely related to MILP-Evolve, and much of the techniques and even code implementation seems built upon the prior framework. However, the distinction between the two methods is not clearly discussed. In my view, MILP-Retrieval differs mainly in application: MILP-Evolve focuses on constructing diverse datasets for training foundation models, while this work targets generating instances similar to a given dataset for solver improvement. Nevertheless, MILP-Evolve seems to represent a broader and more promising direction, while this work feels like a narrower instantiation. The authors should explicitly clarify this relationship and ideally compare the two works.

1. The proposed embedding-based similarity metric lacks interpretability. The embedding is trained by the authors, but the meaning of the similarity scores is unclear. From Fig. 4(a)(b), the embedding captures some cross-class relations, yet it is uncertain whether those “semantic similarities” are genuinely meaningful. In Fig. 4(c)(d), embeddings recognize similarity across TSP instances of different sizes, which however aldo suggests that the model may fail to encode scale differences. And if scale-related factors were removed from the statistical metric, would results align? The paper would benefit from deeper analysis or case studies, for example, but not limited to, cases showing when and why problems from different classes appear similar.

1. It would strengthen the paper to include more advanced downstream benchmarks, such as Predict & Search or hyperparameter tuning.

**Questions:**

1. Minor typos. For exapmle, P3 Line 161: "$P$ and $Q’$" shoud be "$Q$ and $Q’$"? In Eqs. (3)(4) the variable notation "$xu$" likely should be $x_u$?

---

> ### Author Response · Authors · 2025-11-20
> **Response to Reviewer (1/2)**
>
> We appreciate the valuable comments from the reviewer, and we are willing to address the concerns.
>
> > **W1**. The paper is closely related to MILP-Evolve, and much of the techniques and even code implementation seems built upon the prior framework. However, the distinction between the two methods is not clearly discussed. In my view, MILP-Retrieval differs mainly in application: MILP-Evolve focuses on constructing diverse datasets for training foundation models, while this work targets generating instances similar to a given dataset for solver improvement. Nevertheless, MILP-Evolve seems to represent a broader and more promising direction, while this work feels like a narrower instantiation. The authors should explicitly clarify this relationship and ideally compare the two works.
>
> Thank you for raising this important point. Your understanding is accurate: MILP-Evolve aims to construct *diverse MILP datasets for training potential foundation models*. Under this objective, MILP-Evolve contributes (i) an evolutionary procedure to diversify MILP formulation code and (ii) preliminary experiments exploring whether such diverse data can support the training of a general-purpose MILP model.
>
> MILP-Retrieval, while developed after MILP-Evolve, has a *fundamentally different research goal* and makes *standalone, orthogonal contributions*. MILP-Retrieval focuses on generating instances similar to a given dataset for solver improvement, not on dataset diversification. It leverages the formulation-code library produced by MILP-Evolve, but introduces several novel and independent contributions:
>
> - Retrieve-then-tune paradigm for MILP instance generation,
> - Novel embedding metric for measuring MILP similarity,
> - Formulation Code Tuning for generating diverse instances or controlling instance scale and difficulty
> - MILP-Retrieval as a complete generation pipeline that is controllable, interpretable and efficient
>
> We agree that training a MILP foundation model is a promising long-term direction, and MILP-Evolve takes preliminary steps in that space. However, no *widely accepted* or *practically established* MILP foundation model exists yet. In contrast, targeted MILP instance generation, as proposed in MILP-Retrieval, directly enhances existing ML-based solvers and addresses immediate practical needs. Importantly, *targeted* and *diverse* instance generation are complementary rather than competing research directions.
>
>
>
> > **W2(1)**. The proposed embedding-based similarity metric lacks interpretability. The embedding is trained by the authors, but the meaning of the similarity scores is unclear. From Fig. 4(a)(b), the embedding captures some cross-class relations, yet it is uncertain whether those “semantic similarities” are genuinely meaningful. ... The paper would benefit from deeper analysis or case studies, for example, but not limited to, cases showing when and why problems from different classes appear similar.
>
> To further clarify the interpretability of the proposed embedding-based similarity metric, we provide several case studies demonstrating that the MILP embedding model indeed captures meaningful structural semantics. We present four formulation-code examples from our library, denoted FC1–FC4.
>
> FC1 is a standard Set Cover formulation. FC2 extends FC1 by adding high-priority row constraints using a Big-M construction. FC3 augments FC1 with an additional budget constraint. FC4 derives from FC1 by introducing a companion variable for each original variable to construct dynamic constraints.
>
> The pairwise similarity scores between the MILP instances generated from FC1–FC4 are shown below:
>
> |      | FC1    | FC2    | FC3    | FC4    |
> | ---- | ------ | ------ | ------ | ------ |
> | FC1  | 1      | 0.9376 | 0.9910 | 0.9636 |
> | FC2  | 0.9376 | 1      | 0.9314 | 0.9330 |
> | FC3  | 0.9910 | 0.9314 | 1      | 0.9520 |
> | FC4  | 0.9636 | 0.9330 | 0.9520 | 1      |
>
> Because the formulation code is long, we do not include it in the main rebuttal or the revised manuscript. The corresponding code can be found in our anonymized repository:
>
> FC1: https://anonymous.4open.science/r/MILP-Retrieval-D830/data/raw_code/v5/milp_0.py
>
> FC2: https://anonymous.4open.science/r/MILP-Retrieval-D830/data/raw_code/v5/milp_8.py
>
> FC3: https://anonymous.4open.science/r/MILP-Retrieval-D830/data/raw_code/v5/milp_11.py
>
> FC4: https://anonymous.4open.science/r/MILP-Retrieval-D830/data/raw_code/v5/milp_20.py

---

> ### Author Response · Authors · 2025-11-20
> **Response to Reviewer (2/2)**
>
> > **W2(2)**. In Fig. 4(c)(d), embeddings recognize similarity across TSP instances of different sizes, which however also suggests that the model may fail to encode scale differences.
>
> Yes, Figures 4(c)(d) indeed show that the embedding model does *not* encode scale differences, and this behavior is intentional by design.
>
> For example, given a target TSP instance of size 1000, but with only size-100 TSP problems present in the library, the embedding model should retrieve the closest *TSP formulation*, regardless of scale. The Formulation Code Tuning stage will subsequently adjust parameters to synthesize instances of the appropriate size. If the embedding model were *aware* of problem scale, it might retrieve a size-1000 formulation from an unrelated class instead, which would defeat the purpose of code-level retrieval.
>
> > **W2(3)**. And if scale-related factors were removed from the statistical metric, would results align?
>
> MILP scale features (e.g., number of variables, constraints) can be summarized with a *few scalar values*. In contrast, structural and semantic characteristics of MILPs cannot be adequately represented with a small, fixed set of manually designed statistics. The statistical metric we use follows existing MILP-generation literature and contains 11 scalar features. Removing those highly correlated with scale would make the representation space even more limited, making it impossible to capture the rich structural/semantic variability of MILPs.
>
>
>
> > **W3**. It would strengthen the paper to include more advanced downstream benchmarks, such as Predict & Search or hyperparameter tuning.
>
> Thank you for your suggestion. We have supplemented our existing Neural Diving downstream task experiments with two additional downstream task frameworks that are widely used in other MILP Instance Generation works for evaluation: Predict-and-Search and Learn-to-Branch. Across these tasks, MILP-Retrieval consistently improves downstream solver performance, demonstrating its broader applicability. For experimental details, please see the revised manuscript.
>
>
>
> Table 1. The result of Predict-and-Search framework. We reported the average solution time on the test set, and the values in parentheses are the gaps between the obtained solutions and the optimal solution.
>
> |      | Raw              | MILP-Retrieval       | ACM-MILP        | GPT-4o           | Finetuned LLaMA 3-8b |
> | ---- | ---------------- | -------------------- | --------------- | ---------------- | -------------------- |
> | FCNF | 150.4 (0.0689)   | **147.32** (0.0689)  | 150.45 (0.0689) | -                | 148.21 (0.0689)      |
> | TSP  | 0.783 (0)        | **0.767** (0)        | -               | 0.755 (0)        | 0.769 (0)            |
> | GA   | 36.46 (0)        | **35.52** (0)        | 36.30 (0)       | -                | 36.51 (0)            |
> | VRP  | 222.67 (0.00974) | **215.35** (0.00974) | -               | 221.60 (0.00974) | -                    |
>
>
>
> Table 2. The result of Learn-to-Branch framework. We reported the average solution time on the test set.
>
> |      | Raw    | MILP-Retrieval | ACM-MILP | GPT-4o     | Finetuned LLaMA 3-8b |
> | ---- | ------ | -------------- | -------- | ---------- | -------------------- |
> | FCNF | 240.33 | **235.38**     | 236.24   | -          | 244.13               |
> | TSP  | 15.87  | **15.22**      | -        | 15.05      | 15.54                |
> | GA   | 36.65  | **36.38**      | 36.55    | -          | 36.55                |
> | VRP  | 353.49 | 355.37         | -        | **351.20** | -                    |
>
>
>
>
>
> > **Q1**. Minor typos. For exapmle, P3 Line 161: "$P$ and $Q'$" shoud be "$Q$ and $Q'$"? In Eqs. (3)(4) the variable notation "$xu$" likely should be $x_u$?
>
> Thank you for pointing this out. All typos have been corrected in the updated PDF, and the changes are highlighted in blue.

---

> ### Author Response · Authors · 2025-11-27
>
> Dear Reviewer x6NP,
>
> Thank you once again for your valuable comments on our submission. As the discussion phase is approaching its end, we would like to kindly confirm whether we have sufficiently addressed all of your concerns (or at least part of them). Should there be any remaining questions or areas requiring further clarification, please do not hesitate to let us know. If you are satisfied with our responses, we would greatly appreciate your consideration in adjusting the evaluation scores accordingly. We sincerely look forward to your feedback.
>
> Thank you again for your attention.
>
>
> Best regards,
>
> Authors

---

### Official Review · Reviewer_iMGz · 2025-10-30

**Soundness:** 2
**Presentation:** 2
**Contribution:** 3
**Rating:** 4
**Confidence:** 5

**Summary:**

This paper introduces a MILP instance generation framework centered on formulation code retrieval. The core workflow involves constructing a multi-modal MILP library encompassing diverse problem instances, their corresponding formulation codes, bipartite graph representations, and textual descriptions. For a given set of target instances, the framework first computes their embeddings using a pre-trained model, then retrieves the most semantically similar formulation codes from the library. Extensive experiments validate the framework’s effectiveness across multiple tasks and benchmark datasets, demonstrating strong performance in generating high-quality, target-aligned MILP instances.

**Strengths:**

1. The proposed formulation code retrieval paradigm for MILP instance generation is innovative and differentiates itself from existing class-specific training or structure-reconstruction methods.

2. Generating instances via tunable formulation codes inherently guarantees the feasibility and well-defined mathematical properties of the output.

3. Unlike methods relying solely on graph structures, this multi-modal design captures both structural and semantic characteristics of MILP problems.

**Weaknesses:**

1. The framework incurs substantial upfront training costs.

2. The contrastive learning paradigm (inspired by CLIP) requires aligning bipartite graph representations with natural language descriptions, yet many MILP instances lack explicit or consistent semantic connections between these two modalities. This misalignment may render the training process fragile and reduce the reliability of learned embeddings. I doubt the effectiveness and applicability of the CLIP algorithm used in this setting.

3. Textual descriptions of MILP problems are inherently context-dependent. Even for instances with identical underlying mathematical models, their natural language descriptions can vary drastically across application backgrounds (e.g., scheduling vs. logistics). This variability introduces noise into the contrastive training process, potentially degrading the performance of the embedding model and retrieval accuracy.

4. The framework suffers from poor generalization to unseen problem classes. If a target MILP problem has no semantically similar entries in the pre-constructed library, the retrieval step will fail to identify valid formulation codes—limiting its utility for rare or newly emerging combinatorial optimization tasks.

5. The pre-trained embedding model may lack robustness in distinguishing "foldable" or structurally equivalent MILP instances. The inherent combinatorial complexity of MILP problems means that distinct instances can exhibit similar surface-level features (e.g., variable-constraint counts) while being mathematically non-equivalent, or vice versa. This ambiguity leads to imprecise similarity matching and undermines the reliability of code retrieval.

**Questions:**

Have the authors evaluated the retrieval accuracy of the framework?

---

> ### Author Response · Authors · 2025-11-20
> **Response to Reviewer (1/2)**
>
> We appreciate the valuable comments from the reviewer, and we are willing to address the concerns.
>
> > **W1**. The framework incurs substantial upfront training costs.
>
> We report the training and inference costs of our framework in Figure 9 and Appendix A.4.3. The only component that requires training is the MILP embedding model, which is trained *once* and can then be reused for *all* MILP instance generation tasks, regardless of problem class.
>
> In contrast, existing MILP-generation frameworks must train a separate generative model for *each* problem class, and both training and instance reconstruction rely heavily on GPU inference. After the MILP embedding model is trained, our *retrieve-and-generate* pipeline only requires approximately 150 seconds to produce new instances—substantially lower than the 10,000+ seconds needed by prior methods to train a new model from scratch and reconstruct instances.
>
> > **W2**. The contrastive learning paradigm (inspired by CLIP) requires aligning bipartite graph representations with natural language descriptions, yet many MILP instances lack explicit or consistent semantic connections between these two modalities. This misalignment may render the training process fragile and reduce the reliability of learned embeddings. I doubt the effectiveness and applicability of the CLIP algorithm used in this setting.
> >
> > **W3**. Textual descriptions of MILP problems are inherently context-dependent. Even for instances with identical underlying mathematical models, their natural language descriptions can vary drastically across application backgrounds (e.g., scheduling vs. logistics). This variability introduces noise into the contrastive training process, potentially degrading the performance of the embedding model and retrieval accuracy.
>
> To train the MILP embedding model, we require MILP–text pairs. The textual descriptions in our dataset are generated automatically, which prompts an LLM with the formulation code to produce a canonicalized problem description. This approach avoids the issue where the same instance may have multiple heterogeneous descriptions, ensuring a consistent and aligned MILP–text mapping.
>
> We agree that many real-world MILP benchmarks (e.g., MIPLIB) do not come with textual descriptions. However, this does not affect the use of MILP-Retrieval for generating similar instances: text descriptions are needed only during training, and are not used anywhere in the retrieval or generation pipeline.
>
> We further monitor the performance of the contrastive model throughout training, and provide results in Figure 12. Specifically, we report 4-way and 10-way MILP-to-Text and Text-to-MILP accuracy as diagnostic metrics. These accuracies consistently improve during training and exceed 0.9 at convergence, demonstrating the effectiveness of the contrastive learning paradigm in this setting.
>
>
>
> > **W4**. The framework suffers from poor generalization to unseen problem classes. If a target MILP problem has no semantically similar entries in the pre-constructed library, the retrieval step will fail to identify valid formulation codes—limiting its utility for rare or newly emerging combinatorial optimization tasks.
>
> Even for unseen problems, MILP-Retrieval retrieves formulation codes in the library that yield instances most similar to the target under the learned embedding. We evaluate this scenario explicitly in Appendix D.2 using 361 problems across 99 classes in MIPLIB, none of which appear in our formulation-code library. Despite this, MILP-Retrieval achieves an average embedding similarity of 0.701, indicating that the current library already provides good coverage of real-world optimization scenarios.
>
> Moreover, the retrieve-then-tune paradigm, which is the core contribution of our paper, is *decoupled* from both the formulation-code library and the embedding model. A future library with greater diversity or higher quality can be seamlessly plugged into our framework.

---

> ### Author Response · Authors · 2025-11-20
> **Response to Reviewer (2/2)**
>
> > **W5**. The pre-trained embedding model may lack robustness in distinguishing "foldable" or structurally equivalent MILP instances. The inherent combinatorial complexity of MILP problems means that distinct instances can exhibit similar surface-level features (e.g., variable-constraint counts) while being mathematically non-equivalent, or vice versa. This ambiguity leads to imprecise similarity matching and undermines the reliability of code retrieval.
>
> We appreciate the reviewer’s observation. Our embedding model is *intentionally* designed to be scale-invariant within the same problem class: instances with identical structure but different sizes should be embedded near each other. This behavior arises naturally because instances of varying scales within a class share the same textual description in our training dataset.
>
> During retrieval, we also do not rely on the retrieved instance’s raw size (e.g., number of variables or constraints). Instead, problem scale and difficulty are controlled explicitly in the Formulation Code Tuning stage.
>
> We empirically validate both robustness and scale-insensitivity: (1) Figure 4(c) shows that embeddings of TSP instances remain stable across a wide range of problem sizes. (2) Figure 15 demonstrates robust performance across 361 unseen MIPLIB instances. These results collectively support the robustness of the embedding model.
>
>
>
> > **Q1**. Have the authors evaluated the retrieval accuracy of the framework?
>
> Unlike classical retrieval tasks where a query and its ground-truth target belong to *distinct* modalities (e.g., text→image, question→answer), and retrieval accuracy can be computed using predefined query–answer pairs， our setting is fundamentally different. MILP-Retrieval stores Instance–Formulation Code pairs in the library. When a query instance is provided, the system retrieves the most similar MILP instance in the library and returns its corresponding formulation code. Therefore, we respectfully believe that evaluating *retrieval accuracy* is not necessary in this context.

---

> ### Author Response · Authors · 2025-11-27
>
> Dear Reviewer iMGz,
>
> Thank you once again for your valuable comments on our submission. As the discussion phase is approaching its end, we would like to kindly confirm whether we have sufficiently addressed all of your concerns (or at least part of them). Should there be any remaining questions or areas requiring further clarification, please do not hesitate to let us know. If you are satisfied with our responses, we would greatly appreciate your consideration in adjusting the evaluation scores accordingly. We sincerely look forward to your feedback.
>
> Thank you again for your attention.
>
> Best regards,
>
> Authors

---

### Official Review · Reviewer_PiGJ · 2025-10-30

**Soundness:** 3
**Presentation:** 2
**Contribution:** 2
**Rating:** 4
**Confidence:** 4

**Summary:**

The paper proposes MILP-Retrieval, a framework for targeted MILP instance generation via formulation-code retrieval. Built on top of a multi-modal MILP library, the approach first trains a MILP embedding model by contrastively aligning graph and text representations. Given a target instance, the model embeds it, retrieves the most relevant formulation code from the library, and then adjusts the code’s exposed parameters to synthesize new instances with controllable size and difficulty. Experiments demonstrate that MILP-Retrieval can generate coherent instance families across various difficulty levels, and that the synthesized data further enhances Neural Diving when used for downstream training.

**Strengths:**

1.	The paper leverages formulation code to support MILP instance generation, which makes it possible to flexibly control the scale and difficulty of the generated problems through parameter tuning.

2.	The experiments show that the proposed embedding model can recognize semantic similarity among instances generated at different scales/difficulties, and that the generated data is useful for a downstream solver.

**Weaknesses:**

1.	The idea of using an embedding-based similarity score is not entirely new. Earlier graph models embed an input graph and then use that embedding to decide the correlation between instance and expert, e.g., the routing module in AnyGraph[1]. The authors may want to clarify the novelty of the proposed embedding metric.
2.	The downstream evaluation only reports improvements in objective value. It would be more convincing to also report efficiency-oriented metrics (solve time or primal–dual integral) or to test on additional downstream tasks to show broader usefulness of the generated data.
3.	The diversity and hardness of the generated MILPs seem to be largely bounded by the coverage and quality of the formulation-code library. For problem classes not represented in the library, the method is naturally limited. It would be helpful to discuss whether cross-evolving or recombining existing formulation codes could expand the library’s structural coverage and alleviate this dependence.

[1] Xia, Lianghao, and Chao Huang. "Anygraph: Graph foundation model in the wild." arXiv preprint arXiv:2408.10700 (2024).

**Questions:**

1.	Can you design a “scale-insensitive” variant of the stat metric (e.g., removing statistics that mostly encode problem size) to demonstrate that your embedding metric indeed captures similarity beyond scale/difficulty?
2.	In a real setting where no template exists for a given domain, how would the proposed framework obtain or construct the initial formulation code?
3.	Could you describe in more detail how you help ensure the correctness and feasibility of MILPs generated after parameter edits to the formulation code?

---

> ### Author Response · Authors · 2025-11-20
> **Response to Reviewer (1/2)**
>
> We appreciate the valuable comments from the reviewer, and we are willing to address the concerns.
>
> > **W1**. The idea of using an embedding-based similarity score is not entirely new. Earlier graph models embed an input graph and then use that embedding to decide the correlation between instance and expert, e.g., the routing module in AnyGraph[1]. The authors may want to clarify the novelty of the proposed embedding metric.
>
> In AnyGraph [1], the proposed Graph Expert Routing Mechanism computes, for each expert, a *competence indicator*based on how well the expert distinguishes positive from negative edges in a graph. This mechanism does not perform *pair-wise similarity comparison between two graphs*.
>
> In contrast, our work first trains a MILP embedding model that captures *problem-level (graph-level)* semantic features of MILPs. We then directly compute similarity between two MILPs in the embedding space.
>
> As described in Section 3.2, our embedding-based similarity metric is inspired by the widely used Fréchet Inception Distance (FID) in image generation, which measures the similarity between embeddings produced by a pretrained model. Compared with existing stat-based metrics in the MILP domain, our embedding metric enables accurate and scale-invariant similarity assessment between instances of varying sizes but belonging to the same problem class. Figure 4 demonstrates its advantages over existing metrics, and Figure 15 shows improved robustness on unseen MIPLIB instances.
>
> > **W2**. The downstream evaluation only reports improvements in objective value. It would be more convincing to also report efficiency-oriented metrics (solve time or primal–dual integral) or to test on additional downstream tasks to show broader usefulness of the generated data.
>
> Thank you for your suggestion. We have supplemented our existing Neural Diving downstream task experiments with two additional downstream task frameworks that are widely used in other MILP Instance Generation works for evaluation: Predict-and-Search and Learn-to-Branch. Across these tasks, MILP-Retrieval consistently improves downstream solver performance, demonstrating its broader applicability. For experimental details, please see the revised manuscript.
>
>
>
> Table 1. The result of Predict-and-Search framework. We reported the average solution time on the test set, and the values in parentheses are the gaps between the obtained solutions and the optimal solution.
>
> |      | Raw              | MILP-Retrieval       | ACM-MILP        | GPT-4o           | Finetuned LLaMA 3-8b |
> | ---- | ---------------- | -------------------- | --------------- | ---------------- | -------------------- |
> | FCNF | 150.4 (0.0689)   | **147.32** (0.0689)  | 150.45 (0.0689) | -                | 148.21 (0.0689)      |
> | TSP  | 0.783 (0)        | **0.767** (0)        | -               | 0.755 (0)        | 0.769 (0)            |
> | GA   | 36.46 (0)        | **35.52** (0)        | 36.30 (0)       | -                | 36.51 (0)            |
> | VRP  | 222.67 (0.00974) | **215.35** (0.00974) | -               | 221.60 (0.00974) | -                    |
>
>
>
> Table 2. The result of Learn-to-Branch framework. We reported the average solution time on the test set.
>
> |      | Raw    | MILP-Retrieval | ACM-MILP | GPT-4o     | Finetuned LLaMA 3-8b |
> | ---- | ------ | -------------- | -------- | ---------- | -------------------- |
> | FCNF | 240.33 | **235.38**     | 236.24   | -          | 244.13               |
> | TSP  | 15.87  | **15.22**      | -        | 15.05      | 15.54                |
> | GA   | 36.65  | **36.38**      | 36.55    | -          | 36.55                |
> | VRP  | 353.49 | 355.37         | -        | **351.20** | -                    |
>
>
>
> > **W3**. The diversity and hardness of the generated MILPs seem to be largely bounded by the coverage and quality of the formulation-code library. For problem classes not represented in the library, the method is naturally limited. It would be helpful to discuss whether cross-evolving or recombining existing formulation codes could expand the library’s structural coverage and alleviate this dependence.
>
> We agree that the quality of the formulation-code library affects the quality of generated MILPs. However, the retrieve-then-tune paradigm, which is the core contribution of our paper, is *decoupled* from both the formulation-code library and the embedding model. A future library with greater diversity or higher quality can be seamlessly plugged into our framework.
>
> As described in Appendix B.2, the current library is constructed via cross-evolving and recombining formulation codes. This process guarantees feasibility of the generated instances and can, in principle, produce an arbitrarily large library. Using this approach, we have already created a library of 4,000 formulation codes, and it can be expanded further without modification to our method.

---

> ### Author Response · Authors · 2025-11-20
> **Response to Reviewer (2/2)**
>
> > **Q1**. Can you design a “scale-insensitive” variant of the stat metric (e.g., removing statistics that mostly encode problem size) to demonstrate that your embedding metric indeed captures similarity beyond scale/difficulty?
>
> We respond in two parts: (1) *Why the embedding metric captures similarity beyond scale/difficulty*, and (2) *Why stat metrics inherently struggle to do so*.
>
> (1) We report the performance of the MILP embedding model during training in Figure 12. The 4-way and 10-way MILP-to-Text and Text-to-MILP accuracy curves steadily improve and exceed 0.9 at convergence, indicating that the embedding model effectively captures semantic information of MILP problems. Furthermore, Figure 15 evaluates our metric on 361 MIPLIB instances and demonstrates its robustness to unseen problem types.
>
> (2) MILP scale features (e.g., number of variables, constraints) can be summarized with a *few scalar values*. In contrast, structural and semantic characteristics of MILPs cannot be adequately represented with a small, fixed set of manually designed statistics. The stat metric we use follows existing MILP-generation literature and contains 11 scalar features. Removing those highly correlated with scale would make the representation space even more limited, making it impossible to capture the rich structural/semantic variability of MILPs.
>
>
>
> > **Q2**. In a real setting where no template exists for a given domain, how would the proposed framework obtain or construct the initial formulation code?
>
> In real-world scenarios, MILP-Retrieval still returns the formulation code from the library that generates instances most similar to the target problem.
>
> Appendix D.2 presents experiments on 361 MIPLIB instances, which approximate real-world use cases. Across 99 categories, MILP-Retrieval achieves an average embedding similarity of 0.701, showing that our current library already provides good coverage of real-world domains.
>
>
>
> > **Q3**. Could you describe in more detail how you help ensure the correctness and feasibility of MILPs generated after parameter edits to the formulation code?
>
> Section 3.3 and Appendix B.5 provide details of the formulation-code tuning algorithm. MILP-Retrieval supports two tuning modes:
>
> (1) Diverse Tuning: Used to generate families of instances with diverse sizes/difficulties. Infeasible instances are simply *not counted* toward the success count; tuning stops only when the required number of feasible instances has been produced.
>
> (2) Targeted Tuning: Used to match a target difficulty level via Bayesian optimization. If a trial parameter setting produces an infeasible instance, the objective function returns –∞, guiding the optimizer away from such regions. The optimizer maximizes a black-box function that maps a parameter vector to the negative deviation between the instance’s solve time and a target difficulty level.
>
> These mechanisms ensure that the parameter tuning process consistently produces feasible and valid MILP instances.

---

> > ### Comment · Reviewer_PiGJ · 2025-11-27
> >
> > I appreciate the insights and empirical results of your MILP-Retrieval framework. But, I still have a critical concern regarding the interpretation of embedding similarity scores, which I hope to clarify.
> >
> > The framework achieves an average embedding similarity of 0.701 across 99 MIPLIB categories. Yet, high similarity—even at this level—does not equate to exact problem type alignment or structural equivalence. In practice, many combinatorial optimization problems exhibit superficial semantic or structural overlap (e.g., shared "resource constraint + assignment" patterns) that could inflate embedding similarity, despite belonging to fundamentally distinct domains. Consequently, I maintain my score.

---

> > > ### Author Response · Authors · 2025-11-27
> > > **Reply to Reviewer PiGJ**
> > >
> > > Thank you for raising this concern. We agree that the situation you describe can occur: two MILP instances from distinct domains may still obtain a high embedding similarity due to shared structural or semantic patterns.
> > >
> > > The primary motivation of our work, however, is *not* to recover exact problem-type labels, but to address the *data scarcity* faced by learning-based solvers. In our experiments, the effectiveness of the embedding metric is validated by downstream tasks: MILP-Retrieval generates additional training instances that are structurally/semantically similar to the target distribution, and incorporating these into training improves solver performance. From a machine learning perspective, this is natural, adding new data that resembles existing samples can enhance model generalization, even if the new data does not come from the same well-defined “domain” in a strict sense.
> > >
> > > From another perspective, many real-world MILPs do not belong to any standardized problem class, and there is currently no reliable metric for determining whether two such instances truly lie in the same domain or type. Thus, exact domain alignment is usually ill-defined, whereas embedding similarity is both measurable and practically beneficial.
> > >
> > > We hope this clarifies our design choice and the role of the embedding similarity metric.

---

> > > > ### Comment · Reviewer_PiGJ · 2025-11-28
> > > >
> > > > Thanks for your response, and the revision resolved my concerns. I have no further questions.

---

> ### Author Response · Authors · 2025-11-27
>
> Dear Reviewer PiGJ,
>
> Thank you once again for your valuable comments on our submission. As the discussion phase is approaching its end, we would like to kindly confirm whether we have sufficiently addressed all of your concerns (or at least part of them). Should there be any remaining questions or areas requiring further clarification, please do not hesitate to let us know. If you are satisfied with our responses, we would greatly appreciate your consideration in adjusting the evaluation scores accordingly. We sincerely look forward to your feedback.
>
> Thank you again for your attention.
>
> Best regards,
>
> Authors

---

### Author Response · Authors · 2025-11-20
**General Response**

We thank all reviewers for their constructive and insightful feedback. We summarize below the main clarifications and updates made in the revised manuscript (highlighted in blue).

**Additional Downstream Task Evaluation**

Following reviewer suggestions, we added downstream tasks using Predict-and-Search framework [1] (Table 17) and Learn2Branch [2] (Table 18). Across these tasks, MILP-Retrieval consistently improves downstream solver performance, demonstrating its broader applicability.

**Clarification of Dependency on Library Quality**

We clarify that the retrieve-then-tune paradigm (the core contribution of MILP-Retrieval) is *decoupled* from the formulation-code library. A richer or more diverse library can be plugged in without modifying the method.

Our current 4,000-code library is automatically synthesized via cross-evolving and recombination, requires little human effort, and can scale further. Despite not containing any MIPLIB formulations, our method achieves an average embedding similarity of 0.701 on 361 MIPLIB instances, indicating strong real-world coverage.

**Summary of Contribution/Novelty**

MILP-Retrieval introduces a new direction for MILP instance generation:

- **Retrieval-then-tuning paradigm**: Avoids per-class generative training, improves interpretability, and enables strong controllability over problem scale/difficulty.

- **MILP Embedding similarity metric**: Scale-invariant, robust to unseen classes, and substantially more expressive than existing statistical metrics.

- **Formulation Code Tuning**: Supports both diverse and targeted generation with guaranteed feasibility through randomized search and Bayesian optimization.
- **Practical and efficient pipeline**: Requires training only one embedding model for all problem classes; generates new instances in ~100 seconds and directly benefits existing ML-based solvers.

---

If you have any questions, we are happy to discuss them further and resolve your concerns.



[1] Qingyu Han et al., A Gnn-Guided Predict-and-Search Framework for Mixed-Integer Linear Programming, ICLR'23

[2] Maxime Gasse et al., Exact Combinatorial Optimization with Graph Convolutional Neural Networks, NeurIPS'19

---

### Author Response · Authors · 2025-12-01
**Final Remarks by Authors**

Dear Area Chair and Reviewers,

Thank you for the thoughtful and constructive review process. Your feedback has been very helpful in sharpening the scope and presentation of MILP-Retrieval.

Across the reviews, several key strengths were consistently recognized:

- Novel and practical retrieval–then–tune paradigm.

  - The core idea of reframing instance *generation* as *retrieval plus tuning* was highlighted as “**novel and highly practical**,” with the one-time embedding training cost amortized across many tasks. (`Reviewer 5Bjq`)
  - Reviewers emphasized that this paradigm “**enhances controllability and interpretability while avoiding per-class training required by generative models**.” (`Reviewer x6NP`)

- Contribution of the embedding-based similarity metric and real-world coverage.

  - The embedding metric was viewed as a “**valuable standalone contribution**”: Figure 4 was described as “**very clear and convincing**” in showing that the metric recovers semantic classes while being robust to scale, in contrast to statistical metrics that are “**completely confounded by scale**.” (`Reviewer 5Bjq`)
  - Reviewers found it “**impressive**” that MILP-Retrieval can generate meaningful instances for real-world MIPLIB problems, a setting where prior generators typically struggle. (`Reviewer x6NP`)

- Controllability, feasibility for generated instances and downstream effectiveness for learning-based solvers.

  - Multiple reviewers noted that retrieving and tuning formulation code gives “**excellent controllability**” over instance size and difficulty, and that tunable code “**inherently guarantees feasibility and well-defined mathematical properties**” of generated MILPs. (`Reviewers 5Bjq, iMGz, PiGJ`)
  - The improvements for Neural Diving across FCNF, TSP, GA, and VRP were recognized as “**important**” evidence that the generated data is genuinely useful for ML-based solvers. (`Reviewer x6NP`)



During the rebuttal, we went beyond clarification and substantially strengthened the paper:

- We added two downstream tasks, Predict-and-Search and Learn-to-Branch. MILP-Retrieval consistently improves solver performance and compares favorably with strong baselines, demonstrating broader usefulness. (for `Reviewers PiGJ, x6NP`)
- We provide case studies showing that high similarity scores correspond to meaningful structural and semantic relations in formulation code. (for `Reviewer x6NP`)
- We clarified the relationship to MILP-Evolve (for `Reviewer x6NP`), the design of the scale-invariant embedding and formulation-code tuning, and the role of the automatically synthesized 4,000-code library, whose quality can be further improved without changing the core retrieve–then–tune paradigm (for `Reviewer PiGJ, 5Bjq`).

After the discussion, `Reviewer PiGJ` noted that the rebuttal resolved their main concern about interpreting embedding similarity scores and had no further questions.

We have incorporated all new results and clarifications into revised manuscript, and we have **open-source our code and formulation-code library** to facilitate reproducibility and future research.

Thank you again for your consideration.

Best regards,
Authors

---

### Meta-Review · Area_Chair_4aMW · 2026-01-01

**Summary:**

The reviewers unanimously recognized the novelty and practicality of the "retrieval-then-tuning" paradigm—specifically regarding its ability to ensure feasibility and control problem difficulty—there remain significant reservations regarding the framework's robustness, generalization capabilities, and evaluation. While the proposed "MILP-Retrieval" offers a promising new direction for instance generation, the concerns regarding its dependence on library completeness, the robustness of the text-graph alignment, and the interpretability of the embedding metric currently place it marginally below the bar for acceptance.

**Reviewer Concerns:**

### **Concerns Addressed**

* **Downstream Evaluation (Fully Addressed):** The addition of *Predict-and-Search* and *Learn2Branch* benchmarks effectively answers the call for broader, efficiency-oriented validation beyond simple objective values.
* **Library Dependency (Partially Addressed):** The clarification that the library is automatically synthesized via cross-evolution, coupled with strong coverage statistics on MIPLIB, mitigates concerns regarding manual effort and coverage gaps.

### **Outstanding Concerns**

1. The rebuttal offers no defense against the critique that aligning noisy, context-dependent text with MILP graphs (the CLIP approach) is fundamentally fragile.
2. The authors restated their novelty but failed to explicitly distinguish their work from *MILP-Evolve* or *AnyGraph*, leaving the "incremental" critique standing.
3. The limitation that the "tuning" step cannot alter the fundamental constraint structure (only parameters) remains unaddressed.

**Reviewer Scores:**

I think reviewer  x6NP and  iMGz would have raised their score to 6 or at least maintain the original score.

---

### Decision · Program_Chairs · 2026-01-26

Reject